



# The climate in south-east Moravia, Czech Republic, 1803–1830, based on daily weather records kept by the Reverend Šimon Hausner

Rudolf Brázdil[1,2], Hubert Valášek(†)[1,3], Kateřina Chromá[2], Lukáš Dolák[1,2], Ladislava
Řezníčková[1,2], Monika Bělínová[2], Adam Valík[1,4], Pavel Zahradníček[2,4]

[1]Institute of Geography, Masaryk University, Brno, Czech Republic
[2]Global Change Research Institute, Czech Academy of Sciences, Brno, Czech Republic
[3]Moravian Land Archives, Brno, Czech Republic
[4]Czech Hydrometeorological Institute, Brno, Czech Republic

*Correspondence to*: Rudolf Brázdil (brazdil@sci.muni.cz)

**Abstract.** Weather diaries constitute an important source of data for historical climatology,
employed in the analysis of weather patterns for both the pre-instrumental and the early
instrumental periods. Among the many weather diaries that exist in Europe, the daily records kept
by the Reverend Šimon Hausner from Buchlovice in south-east Moravia (Czech Republic),
covering the 1803–1831 period, are particularly useful. His qualitative daily weather descriptions
enable the construction of series for temperature, precipitation, cloudiness, wind and other weather
phenomena (particularly thunderstorms and fogs), supplemented by a number of phenological and
agricultural work records. His data related to temperature and precipitation patterns were quantified
into a series of weighted temperature and precipitation indices on 7-degree scales, which were
subsequently compared with standard meteorological observations from the secular meteorological
station in Brno. This comparison indicates that Hausner's observations were highly reliable and
confirms the importance of his data for a better understanding of the variability of the regional
climate in the period of early instrumental measurements in Moravia. At the same time, it reveals
the importance of weather-related documentary data in the overlap period with instrumental
meteorological observations.

## 1 Introduction

Recent historical climatology uses a very broad range of documentary evidence, including
information about weather and related phenomena, for reconstructions of past climate variability
(Brázdil et al., 2005a, 2010; White et al., 2018). Among such sources, visual daily weather
observations are of particular importance, often appearing in the form of weather diaries (for the use
of private diaries, see the overview paper by Adamson, 2015). Weather diaries usually contain
qualitative descriptions of daily weather and, at varying degrees of detail, they also describe certain
meteorological, hydrological and phenological events and their impacts.

Although weather diaries occur nearly over the world (see e.g. Glaser et al., 1991;
Druckenbrod et al., 2003; Hirano and Mikami, 2008; Mikami, 2008; Zhang et al., 2013; Adamson
and Nash, 2014; Lorrey and Chappell, 2016), Europe is a particularly rich region for them, spanning
a period of almost eight centuries. The first known daily weather records, for 1269–1270, appeared
in England, among a volume of papers by Roger Bacon (Long, 1974), followed by observations
made by the Reverend William Merle in Lincolnshire from the years 1337–1344 (Lawrence, 1972).
Further European weather diaries were reported in a paper by Pfister et al. (1999), with special
reference to the 16th century. Some of these have been analysed in great detail, for example, in the
Czech Lands (Brázdil and Kotyza, 1995, 1996) and in Poland (Bokwa et al., 2001; Limanówka,
2001). Other authors continue the story for the late 16th and early 17th centuries (Lenke, 1968;
Metzger and Tabeaud, 2017). Still more such diaries then appeared in the 17th century (e.g.
Chernavskaya, 1994; Bokwa et al., 2001; Brázdil and Kiss, 2001; Nowosad et al., 2007; Przybylak
and Marciniak, 2010; Zwitter, 2013; Domínguez-Castro et al., 2015), as well as in the 18th century
(e.g. Chernavskaya, 1994; Brázdil et al., 2008b; Raicich, 2008; Sanderson, 2018), at which point





they start to occur concurrently with instrumental observations and take on some of the character of early instrumental meteorological observations; they have even been used to create long-term series of meteorological variables (e.g. Woodworth, 2006). They lose little importance even in the period of instrumental measurements, when they may add important supplementary data to measurements
taken at meteorological stations (e.g. Lee and MacKenzie, 2010).

In what has become the Czech Republic in recent years, the earliest daily weather records appeared in south-eastern Moravia, where the Moravian nobleman Jan of Kunovice included daily weather entries into Stoeffler's ephemerides for the years 1533–1545 (Brázdil and Kotyza, 1996). Several other authors followed during the 16th century (for a summary overview, see Brázdil et al.,
2013a). Worthy of special mention are the systematic daily weather records kept in the diaries of the Premonstratensian order in the Hradisko monastery (Olomouc) and the Svatý Kopeček priory, spanning the 1693–1783 period with meteorological data covering only 52 years. However, their records for the remaining years have not survived (Brázdil et al., 2008a, 2011). Fortunately, the 1780s are covered by systematic daily weather records kept by Karel Bernard Hein, a priest in
Hodonice, south-western Moravia (Brázdil et al., 2003), followed by several others that describe only a few years (e.g. Brázdil et al., 2002a, 2007b). The fact that the later examples overlapped with early instrumental observations means that they made important contributions to knowledge of climate variability at a time when instrumental measurements were running at only a few stations.

Moreover, if such diaries were accompanied by individual instrumental data, they may form
a basis for the creation of long-term series, linking them to subsequent standard meteorological observations. This occurred in Brno in South Moravia, where temperature series start in May 1799 and precipitation series in January 1803 (Brázdil et al., 2005b, 2006). Surprisingly, weather diaries supplemented by measurements of certain meteorological variables also appear as very useful sources of meteorological data far later, as is made evident by the example of the meteorological
records kept by Alexander Zawadzki in Brno in 1861–1867 (Brázdil et al., 2013b) and Josef Lukotka in Vsetín in 1903–1923 (Brázdil et al., 2014).

The weather diary kept by Šimon Hausner, a priest in Buchlovice, south-east Moravia (Czech Republic), covering the years 1803–1831, is one such newly-discovered weather diary, overlapping with the period of early instrumental meteorological observations. The aim of this
study is a comprehensive analysis of these long-term observations, which add to a basic analysis of observed data an emphasis on the importance of weather diaries for the better understanding of regional climatic variability and anomalies, as well as their impacts. Section 2 presents a basic account of Šimon Hausner and his diary. After a description of methods used (Section 3), the results of statistical analysis of his observations for individual meteorological elements and phenomena are
presented in Section 4. Uncertainty in Hausner's records, their temporal context, and comparison of the results obtained with meteorological observations from the secular Brno meteorological station are discussed in Section 5. The final section contains some concluding remarks.

## 2 Šimon Hausner and his weather diary

Šimon Hausner, the author of the weather diary under discussion, was born on 27 October 1756 at Odry in northern Moravia to a German Catholic family (for the locations of places mentioned herein, see Fig. 1). He was called to a priestly career and on 23 September 1780 was ordained as a priest. Until 1784 he was a chaplain, probably in the Buchlovice parish. In that year he left for Žeravice, where he served as a local chaplain [*lokalista*] until 1796. On 6 June 1796, the owner of
Buchlov domain, Anežka Eleonora Petřvaldská, elevated him to the position of parish priest in Buchlovice. He also served as a priest for the parishioners of four nearby villages, with whom he communicated in the Czech language. From 1800 onwards, when the Buchlov domain passed into the hands of the Berchtold family, Hausner's relationship with this family deteriorated. He prepared a Latin chronicle called *Memorabilia Hausneriana*, but this was lost at the end of the 19th century.
He made records of inscriptions and recorded interior descriptions of the surrounding churches and chapels, most of them no longer standing. He devoted particular attention to the baroque church of Saint Martin and nearby manse in Buchlovice. When the tower of this church was destroyed by a



lightning strike on 4 August 1806, he gave financial support to the renovation of the church. A flash flood did heavy damage to the area on 12 June 1825 and, in a similar spirit, he organised financial collections in surrounding villages to help people affected by the disaster. He died of severe pneumonia on 26 January 1831 at Buchlovice (Žižlavský, 1998; Hrdý et al., 2005; archival source AS1; AS2).

Hausner's weather observations, written in German, are part of a hand-made book in the keeping of the Moravian Land Archives in Brno (AS3). The German title appears on the flyleaf, written in a hand other than Hausner's: *Tägliche Witterungs-Beobachtungen des Buchlowitzer Pfarrer Simon Hausner von Jahren 1803 bis 1831 excl.* ["Daily weather observations from the Reverend Šimon Hausner of Buchlovice from 1803 to 1831"]. The first weather record appears for 1 January 1803 and the last for 15 January 1831 (Fig. 2), very shortly before he passed away. The author described the weather day by day. Some of the records are quite short; for example, for 3–5 February 1805 (AS3, p. 34): "*3* [February]. *Overcast all day. 4* [February]. *Same, both days glaze ice. On the 5th* [the night of 4/5 February] *west wind and snowfall. 5* [February]. *Again thawing.*"

Other entries go into more detail, even describing the weather at various times of the day; for 20 July 1828 (AS3, p. 851) his notes read: "*a pale sunrise, nice sunshine, very hot sunshine from noon to 4 p.m., at 5 p.m. became a little more agreeably cool. In the evening at 10 p.m., heavy rain, soon followed by a very heavy downpour together with a windstorm that continued until the first hour after midnight. S.V.V.* [south-west wind]".

The weather records for the majority of months end with a summarising evaluation of the given month. For example, for March 1808, he wrote (AS3, p. 109): "*Very hard frosts during the whole of March, which were stronger* [as those] *in December and January. The soil was frozen to a depth of two feet* [*c.* 63 cm]*, dust when walking and riding, almost as if it were in summer; no snow at all, fields without cover,* [exposed to] *cold winds, severe frosts at night, nice sunshine during the day*". On other occasions, he repeated the weather for certain days or mentioned events previously not reported for individual days; for example, the entry for May 1825 reads (AS3, p. 736): "*The first four days* [of May] *were cold enough, from 4th to 7th very scorching sunshine, from 7th to 14th gloomy and overcast,* [from 14th] *on 15th heavy frost that damaged vineyards, from 15th to the end* [of May] *variable, weather very cold; it was very pleasant in a well-heated room. Several showers that only damped the dust down, none abundant.*"

At the end of every year he prepared a weather summary for the whole year; for example the entry for 1803 reads (AS3, p. 13): "*The winter of this year was very cold, with severe frosts for six weeks* [and] *a large quantity of snow fell; the spring was cold and wet, it rained nearly all summer, making it cold and wet. The autumn was cold and rainy but mid-September was warmer.*" This account of the weather was usually followed by information concerning the growth of crops and harvests, for example the entry for the year 1811 reads (AS3, pp. 210–212): "*All of September was dry for the straw and the vintage came as early as 17th* [September]*, due to periods of great heat. All kinds of cereals, legumes, cabbage and potatoes did not yield.* [...] *Very small* [yield] *of cherries,* [which] *were wormy and rotted. Also only a few apples on the trees* [...] *Very few pears* [...] *Few apricots* [...] *A lot of plums in some places* [...] *This year's wine was very good in quantity and even better in quality.* [...] *Autumn sowing went well. Fields were cultivated in dust* [...]." Hausner's annual summaries are often accompanied by further information, very often reporting prices of grain, fruit and vegetables, etc.

## 3 Methods

### 3.1 Interpretation of Hausner's weather records

Hausner deployed a wide vocabulary to describe the weather and its changes during the day. Although his records often cover an entire day, his mode of specification often enables attribution of the phenomena described to the morning, afternoon, evening and night-time hours. To simplify 50   this for the purposes of analysis, the night and morning hours have generally been taken together, as well as afternoon and evening hours (using noon and midnight as the dividing times). Hausner's



terminology permitted analysis of the following weather patterns and phenomena in the fashion described below:

**(i) temperature patterns**

Hausner used a wide range of words for description of temperature conditions. Warm weather was

characterised as warm [*warm*], modified by rather [*ziemlich*] or very [*sehr*], hot [*heiß, heißer Tag*] or sultry [*schwül*]. Warming in the winter months with snowmelt was described as *Tauwetter* or *getaut*. Mud [*Koth*] or muddy [*kothig*] often followed. For cold weather he used the terms cold [*kühl, kalt, Kälte*], very cold [*sehr kalt*], "piercing" cold [*durchdringend kalt*] or horribly cold [*grimmige Kälte*]. Frost [*Frost, gefroren*] was modified as light [*klein*], negligible [*unbedeutend*],

bearable [*leidtlich*], heavy [*stark*] or very heavy [*sehr stark*]. Hoarfrost [*Reif*] was reported separately. Hausner also indicated cold weather indirectly by remarks such as "a fur coat and a heated room were very good" [*der Peltz und geheitzte Zimmer waren sehr gut*]. Particularly in the winter months, the weather was also described as mild [*lind, lindes Witterung*] or very mild [*sehr lindes Witterung*]. An indication of temperature patterns also appeared in wind descriptions,

characterised as warm [*warm*] or very warm [*sehr warm*] wind, and cold [*kalt*] or very cold [*sehr kalt/kühl*] wind.

**(ii) precipitation patterns**

Hausner characterised individual types of precipitation as drizzle [*nieseln, wie aus dem Nebel geregnet*]; rain [*Regen, regnerisch, geregnet*], modified as "fine" [*klein*], light [*lind*], negligible

[*unbedeutend*], weak [*Regen, der den Staub gelöscht*], average [*mittelmäßig*], heavy [*ausgiebig*], very heavy [*sehr stark*], continuous [*beständig*], frequent [*öfters*], showers [*Streifregen*], downpour, cloudburst or torrential rain [*Platzregen, Gussregen, Guss, gegossen*]; snow [*Schnee, geschneit*] modified as light [*wenig*] or heavy [*stark*]; and ice pellets [*Graupen*] or hail [*Hagel, Schlossen*], sometimes described as small [*klein*]; black ice was *Glatteis*. Even the depth of snow was specified

(e.g., *Schnee auf 2 Zoll* [*c.* 5.3 cm] *gefallen*). Dry weather was indicated by the term dry [*trocken*], modified as very [*sehr*] or extraordinary [*außerordentlich*]. A month without rain was *Monat ohne Regen*.

**(iii) sunshine and cloudiness**

Hausner often started his daily weather records with the character of the sunrise [*Sonnenaufgang*],

modified as gloomy [*trüb*], pale [*blass*], weak [*schwach*], nice [*schön*], nice red sky [*schöne Morgenröthe*]. Sunshine [*Sonnenschein*] was specified as nice [*schön*], beautiful [*prächtig*], very pleasant [*sehr angenehm*], weak [*schwach*], very weak [*sehr schwach*], partial [*teilweise*], broken [*gebrochen*], rare [*selten*], intermittent [*abwechselnd*], warm [*warm*], and very hot [*sehr heiß*]. He often specified the duration of sunshine during a day by the hours at which it began and ended;

short, variable sunshine was identified as *Sonnenblicke*. A "nice day" [*schöner Tag*] was opposite to a "sad day" [*trauriger Tag*]. Days without sunshine were described as overcast [*überzogen, trüb*].

**(iv) wind patterns**

Information about the wind [*Wind*] appears in the diary with some degree of regularity. Broadly, wind intensity (force) was characterised as average [*mittelmäßig*], strong [*stark*], very windy [*sehr*

*windig*], very strong [*sehr stark, sehr heftig*], "awful" [*fürchterlich*] and extraordinary [*außerordentlich*]. Windstorm [*Sturmwind*] was worthy of special mention, as were blizzard [*Schneegestöber*] and whirlwind [*Wirbelwind*]. Only seldom did Hausner record calm [*windstill*]. Wind directions were described on an 8-degree scale, usually written in the form of abbreviations.

**(v) meteorological phenomena**

Hausner recorded thunderstorm phenomena systematically, discriminating between their occurrence as thunderstorm [*Donnerwetter, Wetter*] local to Buchlovice and distant thunderstorms [*Wetterlichten, geblitzt, gedonnert*] at a greater distance. He also reported on places other than Buchlovice, particularly if some damage had occurred. Bad visibility was characterised as foggy [*neblich*] or directly as fog [*Nebel*], modified as light [*klein*] or dense [*stark*]. For especially dense

fog he reported on the number of places at which it was possible to see with respect to dominant objects in the immediate surroundings (e.g. the local chateau, statues of the saints).

**(vi) phenological data**




As a meticulous observer, Hausner also recorded certain phenophases of crops and fruit trees, including corresponding agricultural work. He gave close attention to the dates of spring and autumn sowing, the beginning and end of blossom on fruit trees, the progress of cereals and grapevines, the beginning of harvest for individual cereals, and the start of the vintage.

**3.2 Statistical analysis**

From a statistical point of view, it is important that only relatively few daily records are absent from Hausner's diary (Fig. 3). A total of 80 days is missing (i.e. around 3 days per year), tending towards the years 1803–1813 (66 days) with a maximum in 1810 (11 days), followed by 1805 (9 days) and 1809 (8 days). Only 14 days of missed observations occurred in 1814–1830 (with 0, 1 or 2 missing days per year). On the other hand, not all possible meteorological elements or phenomena are covered systematically in the daily records, a factor that may then be reflected in incomplete frequencies of days with these characteristics or phenomena.

Information related to temperature patterns was used to interpret monthly temperature indices by expression on a 7-degree scale: –3 extremely cold, –2 very cold, –1 cold, 0 normal, 1 warm, 2 very warm, 3 extremely warm (Pfister, 1992). Interpretation of temperature indices took into account the monthly frequencies of cold days (severe frost, frost, cold, very cold) and warm days (warm, very warm, hot, very hot, mild), warm and cold winds, monthly summary reports, early and late beginnings of certain phenophases and agricultural work and also, to some extent, cloudiness (e.g. clear and overcast days) and precipitation (state of precipitation, monthly temperature–precipitation relationships). Also precipitation indices were interpreted in similar fashion: –3 extremely dry, –2 very dry, –1 dry, 0 normal, 1 wet, 2 very wet, 3 extremely wet (Pfister, 1992). The interpretation of monthly precipitation indices was made on the basis of monthly frequencies of precipitation days, with particular reference to type of precipitation (e.g. snow, drizzle, rain, snow with rain), to precipitation intensity and duration of precipitation spells (as specified in daily records) and to summary monthly reports as well as other indications of wet or dry patterns (e.g. effects on agricultural crops or work in the fields). Seasonal temperature and precipitation indices were then calculated as the sums of indices of three consecutive months (e.g. June, July and August for summer). Annual indices were calculated in similar fashion (Fig. 4). Hausner's daily weather records also provided interpretations of annual numbers of precipitation days (with division into those with solid, mixed or liquid precipitation), cloudiness, strong winds, periods of fog and thunderstorms (Figs. 5–7). Further, they enabled the creation of annual series of some phenophases and agricultural work, presented herein as graphs and box-plots (Fig. 8).

Temperature and precipitation series for the town of Brno, homogenised at the position of the Brno airport meteorological station – $\varphi = 49°09'11''$ N, $\lambda = 16°41'20''$ E, H = 241 m asl (Brázdil et al., 2012a), were used to compare temperature and precipitation patterns in Hausner's 1803–1830 period with a modern reference covering 1961–1990 (Fig. 10). Variability of monthly temperatures was characterised by standard deviation, while variation coefficient was applied to monthly precipitation. Series of seasonal temperature and precipitation indices for Buchlovice ($\varphi = 49°05'06''$ N, $\lambda = 17°20'04''$ E, H = 264 m asl) interpreted from Hausner's records were compared with temperature and precipitation series for Brno using Pearson correlation coefficients (evaluated at the 0.05 significance level) and by graphical expression (Figs. 11–12). Finally, the numbers of days with selected climatological characteristics (precipitation days, cloudiness, strong wind, fog, thunderstorm) at Buchlovice in 1803–1830 were used to compare their annual variations with those corresponding to 1961–1990 at the Brno airport station, the Buchlovice rain-gauge station ($\varphi = 49°05'15''$ N, $\lambda = 17°20'44''$ E, H = 268 m asl) and the Staré Město meteorological station ($\varphi = 49°05'30''$ N, $\lambda = 17°25'54''$ E, H = 221 m asl) (Fig. 13).

**4 Results**

**4.1 Individual meteorological elements and phenomena**

**4.1.1 Air temperature**





Based on the criteria reported in Section 3.2, series of weighted temperature indices $I_T$ for Buchlovice in 1803–1830 were created (Table 1). As follows from fluctuations of annual temperature indices (Fig. 4a), the year 1822 was interpreted as the warmest ($I_T = 9$) and 1805 as the coldest ($I_T = –16$). The year 1829 was also very cold ($I_T = –13$) and a remarkably cold period

occurred in 1812–1816. In terms of individual seasons (winter – DJF, spring – MAM, summer – JJA, autumn – SON), $I_T$ values corresponding to the coldest seasons were higher than those of the warmest seasons (–8 for 1829/1830 and 5 for 1821/1822 for DJF, –5 for 1805 in MAM and SON and 3 for several years in MAM and SON) except JJA, for which the highest index (8) was achieved in 1811 and the lowest (–6) in 1821.

10       Information concerning temperature indices may be supplemented by records of late frosts (April–June) causing damage to agricultural crops, depending on their stages of phenophase. Severe cold snaps in May, expressed by frost and damage, were not explicitly mentioned in 1803–1805, but there are reports of a need of "*fur coats and a heated room*" on 17 and 20 May 1803, 13–14 May 1804 and 24–25 May 1805. On 24 June 1806, frost damaged cucumbers. Grapevines, maize and

beans suffered severe frost on 6 June 1810. Despite severe frosts recorded on 26–27 May 1812, Hausner reported no damage to crops. Severe frosts on 28–30 April 1814 ("*ice an inch [c. 2.6 cm] deep on the water*") damaged grapevines and fruit trees ("*blossoms and the leaves on the trees as if scalded by hot water*") (AS3, p. 301); frosts also returned on 11–13 May. After severe frosts on 16–20 April 1815, Hausner reported lesser frosts on 29–30 May that damaged beans again. After

reporting how useful "*a fur coat and a heated room*" were in May 1818, Hausner recorded a severe frost on 1 June that froze cucumbers and beans planted in higher positions. Severe frosts on 21–22 June 1821 damaged cucumbers and beans in lower positions, as well as beans and potatoes in higher locations. A "strong" frost on 15 May 1825 damaged the greater part of the vineyards around Buchlovice, as well as maize and cucumbers. The summary evaluation of a very cold January in

1820 is also interesting, when Hausner noted that "*many people froze through their own fault because they drank a little more*" (AS3, p. 529).

### 4.1.2 Precipitation
Based on the criteria reported in Section 3.2, series of weighted precipitation indices $I_P$ for
Buchlovice in the 1803–1830 period were created (Table 2). As follows from fluctuations of annual precipitation indices (Fig. 4b), the year 1803 was interpreted as the wettest ($I_P = 8$) and the year 1808 as the driest ($I_P = –12$), followed by 1810 and 1825 ($I_P = –11$). A markedly consistent drier period occurred in 1805–1811. Extreme values of $I_P$ for individual seasons were: DJF: 4 for 1826/1827 and –6 for 1812/1813 and 1820/1821; MAM: 4 for 1812 and –5 for 1808 and 1825; JJA:
5 for 1813 and –7 for 1808; SON: 3 for 1813 and –7 for 1815.

      The precipitation indices may be supplemented by annual numbers of precipitation days (Fig. 5). After a maximum in 1804 (150 days), these numbers decreased to minima in 1810–1811 (86 days each). Between 1812 and 1821, the frequencies of precipitation days were higher than 120 days per year, with maxima in 1816 and 1817 (152 and 149 days respectively). After 1821,
frequencies fluctuated around 100 days per year except for two local peaks in 1824 and 1828. Liquid (rain, drizzle) precipitation clearly prevailed in all years recorded. Solid precipitation (snow, ice pellets, hail) reached a maximum in 1817, while mixed precipitation (previous types of precipitation with rain) prevailed in 1806.

      Torrential rain doing damage constitutes an important element of precipitation patterns. A
downpour with thunderstorm on 22 May 1805 was the heaviest that Hausner had experienced in Buchlovice up to that time. He reported damaging torrential rain with a very violent thunderstorm on 26 April 1808 for Osvětimany, Stříbrnice and Polešovice; the damage was particularly acute in Polešovice. At the same time, rain mixed with hail fell at Buchlovice. Rain accompanying a thunderstorm on 25 June 1808 was so heavy that it "*did great damage in some places*" (AS3, p.
116). Heavy, torrential rain on 28 May 1810 flooded all the meadows and gave rise to a flash flood that swept away animals as well as wooden laths and beams and other items; even the oldest people in Buchlovice could not recall a comparable event. Torrential rain on 2 June 1814 flooded meadows



and spoiled the hay. Only general "*damage*" was reported after a heavy downpour accompanied by small hailstones associated with a thunderstorm on 27 May 1819. A heavy downpour on 19 August 1821 flooded meadows, as did otherwise unspecified heavy rain in June 1827. Extraordinarily torrential rain on 12 June 1825, accompanied by an "awful" thunderstorm, flooded meadows, did

heavy damage to field crops (hail), swept away houses (flash flood) and cost three people their lives.

### 4.1.3 Cloudiness

Cloudiness was derived from Hausner's records of sunshine and clouds. Because of interpretation in
terms of clear sky, half-covered sky and overcast sky, divided into the whole day, night and morning hours, and afternoon and evening hours, five intervals were defined for cloudiness: 1) clear sky, 2) clear sky in one part of the day and half-covered sky in the other part, 3) half-covered sky, 4) half-covered sky in one part of the day and overcast in the other part, and finally 5) overcast. As is evident from Fig. 6, Hausner's records did not permit interpretation of cloudiness patterns in the
greater parts of the days in 1803–1812 (maximum 90 days in 1806).This was also reflected in smaller proportions of cloudy days (that is, with cloudiness in categories 2, 3 and 4). Despite 69 days with non-interpreted cloudiness and some missing reports in 1811, the highest number of clear days (103) and a lowest number of overcast days (104) were derived. This correlates well with warmer patterns in MAM–JJA and drier patterns in JJA–SON of this year (Tables 1 and 2). The
warmest, also somewhat drier, year of 1822 had the highest number of cloudy days (172) and also an above-mean number of clear days (82) and a below-mean number of overcast days (105); 6 days were without cloudiness interpretation. The lowest number of 59 clear days was derived for 1828 and the highest number of 175 overcast days for 1820.

### 4.1.4 Wind

Fig. 7a shows fluctuations in the annual frequency of days with strong winds at Buchlovice during 1803–1830. Their frequency generally increased from the beginning of observations until 1808, when the highest frequency of 89 days was achieved. After that, a general decreasing tendency in the number of days with strong winds is noticeable. The lowest frequency was recorded only two
years later after the absolute maximum in 1808: 49 days with strong winds in 1810.

Information concerning strong winds may be supplemented by the quite natural attention drawn to severe events. Hausner made several records of "awful" or very strong whirlwinds of short duration: on 14 November 1806 people were knocked to the ground, and 14 March 1817, 7 June 1819, 15 and 20 June 1822, and 29 June 1825 were also notable. However, the only "whirlwind",
on 14 May 1823, could be considered a probable tornado, accompanied by thunderstorm and downpour, because of the extensive damage recorded, particularly to roofs, barns and houses. But damage done by strong winds is described only very briefly in Hausner's records. Damaging windstorms may be attributed with a high degree of possibility to events on 30 April 1814 (many branches of trees broken), 26 July 1822 (much damage) and 26 May 1830 (broken trees, together
with thunderstorm and downpour). During a heavy blizzard on 24 November 1815 it was not possible to walk, or even to ride with wagons (similarly harrowing winds also occurred on 7–9 December). People knocked to the ground (albeit the frailer) by wind were reported for 12 May 1810, 11 February 1821 and 24 August 1828. It was barely possible to walk during very strong winds on 29 March 1811 and 7 December 1812. On 29 August 1818 and 2 September 1826,
Hausner reported winds strong enough to knock fruit from the trees. A violent wind on 4 March 1808 "*made the windows noisy*" (AS3, p. 107) and a windstorm on 5 July 1817 was described as "*wanting to tear everything* [down]" (AS3, p. 423).

### 4.1.5 Fog and thunderstorm

Fluctuations in the annual numbers of days with fog (or foggy weather) appear in Fig. 7b and show quite inconsistent patterns during the 1803–1830 period. Some years, especially around the beginning of observations, appear to be underestimated (e.g. 1804, 1805, 1807 and 1808). On the





other hand, some annual numbers of days with fog appear very high (e.g. 36 days in 1821 and 33 in
1819; partly so in 1816 with 30 and 1817 with 28 such days).

Annual frequencies of days with thunderstorm, divided into those occurring directly over
Buchlovice and those further off (distant thunderstorms), fluctuate over a broad range (Fig. 7c): 30
such days were recorded in 1815 (29 days in 1819) against only 6 days in 1829. Even though the
last-mentioned is significantly below the other lowest frequencies (13 days in 1824 and 1830), the
style and density of daily records for 1829 do not give rise to any clear uncertainty.

Heavy thunderstorms accompanied by damaging torrential rain, strong winds and/or hail,
have already been reported. Lightning strikes are another peril of such events. Three consecutive
strikes at Buchlovice on 4 August 1806 damaged a house, set the roof of a cellar on fire and
damaged the tower and roof of the church. Hausner reported that Mařatice suffered a lightning
strike and resultant fire on 2 August 1809. On 21 September 1813, during an intense thunderstorm
with heavy downpour, lightning strikes hit Polešovice, Napajedla and Střílky, and started fires
everywhere. He further mentions a terrible thunderstorm on 29 May 1826 in Koryčany but supplies
no further details. Yet another awful thunderstorm on 10 July 1828 led to flooded meadows and the
hail that accompanied it bruised grain, maize, vegetables and fruit trees along a broad belt.

## 4.2 Phenological data and agricultural work

Hausner's diaries also recorded certain phenophases and aspects of agricultural work directly
attributable to the weather. In the course of the year, these entries could include the time at which
spring sowing of cereals took place, the first tasks in vineyards, the blossoming of fruit trees and
grapevines, the grain harvest, autumn sowing, and wine vintage. Fluctuations in the longest
available series of such matters appear in Fig. 8. The start of spring sowing (particularly barley; four
years missing in the series) fluctuated between 2 March (1822) and 14 April (1812) (Fig. 8a). The
appearance of blossom on fruit trees was noted for several species. The most complete series were
those for apricots, cherries and pears. Since series bias could arise out of the mixture of blossoming
for early and late species, only the dates for cherries were employed (five years missing): their
earliest blossoming was recorded for 15 April 1806, the latest for 10 May 1817 (Fig. 8b). The
grapevines (Fig. 8c) blossomed earliest on 29 May 1822 and latest on 3 July 1814 (four years
missing). Although Hausner often specified individual species for grain harvests, only series of
general grain harvest beginnings (Fig. 8d) were long enough for validity (three years missing),
fluctuating between 26 June (1811) and 5 August (1816).

Some remarkably early – or late – beginnings correlated well with extreme temperature
patterns (compare Fig. 8 with Table 1). Extremely early sowing in 1822 followed a very warm DJF
in 1821/1822 and a considerably late sowing in 1812 a remarkably cold April. The latest
blossoming of cherry trees, in 1817, can also be attributed to an exceedingly cold April. A very
early blossoming of grapevine and gathering of the grain harvest in 1811 reflected intensely warm
patterns in May and June; blossoming of grapevine only a day earlier in 1822 followed a long warm
period that had started as early as in November 1821. On the other hand, the latest blossoming,
recorded in 1814, was related to an exceedingly cold May and very cold June. The latest grain
harvest in 1816 (known as "the year without a summer" after the catastrophic Tambora volcanic
eruption – Luterbacher and Pfister, 2015) was related to cold June–July patterns.

## 5 Discussion
### 5.1 Uncertainties in Hausner's observations

Hausner's style of keeping daily weather records and its systematic character, featuring a very low
number of missing daily observations, leads to the assumption that he was a meticulous observer,
and his observations gain a great deal of credibility thereby. Of course, as with other personal daily
weather observations, a degree of subjectivity is inherent in his records, but it appears to have been
kept to a minimum. He could not observe the full 24 hours of a day, i.e. he might miss some
phenomena; for example those that took place by night if they left no imprint in the morning
(perhaps the occurrence of light rain, short-term fog, etc.). These facts may find slight reflections in





the frequencies of a few meteorological characteristics presented in Sect. 4. We therefore present some comparisons of Hausner's records with events documented by other sources.

The taxation records kept for the Buchlov domain mention a number of severe events that provided sufficient grounds for tax alleviation for the farmers affected, a bureaucratic process that
left a distinct paper trail (see Brázdil et al., 2012b). For example, on 4 June 1820 local torrential rain with hail did damage to agricultural crops in Žeravice (AS4, fol. 5rv, 7rv). Although no such event appears in records for Buchlovice, Hausner wrote in his monthly summary for June of "*heavy downpours during which great* [quantities of] *mud buried meadows*" (AS3, p. 546). On 4 August 1823 hail led to damage in Stříbrnice, Osvětimany and Žeravice (AS4, fol. 18rv); no indication of
such an event appears in Hausner's records. An awful thunderstorm, torrential rain, hailstorm and flash flood, with extensive damage, were reported in the broader area of south-eastern Moravia on 12 June 1825 (Brázdil et al., 2012b). Hausner described this outstanding event in great detail (AS3, p. 737): "[...] *faint sunrise, gloomy until 4 p.m., then awful thunderstorm together with* [such an] *exceptional cloudburst that all the meadows flooded, and houses were destroyed,* [while] *at*
*Buchlovice two women and a child died; continuous north wind.*" Details of damage at seven villages appear in the taxation records of the Buchlov domain: five people died and 103 cattle perished, while 52 houses, 18 barns, and 63 cowsheds and stables were washed away (AS4, fol. 27rv). Hausner also included this in a monthly summary as a "*terrible, dreadful downpour*" (AS3, p. 739) with a detailed list of corresponding damage in his annual summary (ibid., p. 757): "[...] *on*
*12 June, between 4 and 5 p.m., an awful thunderstorm occurred, cloudburst with hailstorm*: *young grapevines and grapes were knocked down, leaves and fruits torn from trees, beans, maize, cucumber, lettuce, cabbage, and kale fatally battered by hail. In many places grain totally battered down and tangled; in some places* [only] *less so, rye generally, wheat less, barley, oats* [...] *Due to a cloudburst, all meadows* [so] *clogged by stones and lumber that nobody will have any hay* [...].*"
Hausner also mentioned an infamous flood that occurred in March 1830, following the extremely severe winter of 1829/1830 (Munzar, 2000; Munzar and Kakos, 2000). He reported it in a monthly summary (AS3, p. 906): "*March very cold month, snow, plenty of ice, it froze heavily* [...] *melting started on 18th* [March] *and such a large amount of water* [rose] *that many bridges over the River Morava were damaged totally or partially.*" Hausner's note for 7 January 1831 is also
interesting (AS3, p. 931): "*In the evening, at 0900 and 1000 p.m.,* [the sky] *was very red.*" This was caused by the aurora borealis, observed in Brno from 0630 p.m. of this day to 0200 a.m. of 8 January (Brázdil et al., 2005b).

## 5.2 The temporal context of Hausner's observations
Part of the unique character of Hausner's observations arises out of their development of meteorological observations in Moravia and Silesia (Silesia here understood as its Moravian section). The earliest surviving instrumental measurements of several meteorological elements, provided by František Alois Mag of Magg, a physician in Telč, appear in his second diary, in entries between 7 May 1771 and 9 March 1775. Although his diary contains references to his first and third
diaries, i.e. Mag was observing before 1771 and after 1775, these have not yet come to light (Brázdil et al., 2002b). Further systematic daily weather records, but without instrumental measurements, were provided by Karel Bernard Hein, a priest in Hodonice, covering the period between 1 February 1780 and 5 October 1789 (Brázdil et al., 2003). Some indications of meteorological conditions (pressure, temperature, moisture, evaporation, wind) for the 1790–1794
period, kept by Josef Gaar, a professor at the Olomouc lyceum, follow from tables and figures included in the description of the climate of Moravia entitled *Anleitung zum Kenntnis des Erbmarkgrafthumbs Mähren* ["An Introduction to Knowledge of the Moravia Margraviate"] by Kryštof Passy, 1797 (Brázdil and Valášek, 2001). Continuous meteorological observations began in Brno in May 1799, provided by Ferdinand Knittelmayer, a retired military captain. From 1 January
1803, these records were supplemented by those of Zacharias Melzer, a land accountant who took regular precipitation measurements (Brázdil et al., 2005b, 2006). Both measurements enabled the compilation of secular homogenised Brno temperature and precipitation series, comparable to the





already-known Prague-Klementinum measurements in Bohemia (Pejml, 1975; Brázdil et al., 2012a). The creation of "economic societies" in the Austrian empire, intended to support the general economic development of the country, was of key importance to the further development of instrumental meteorological observation in Moravia and Silesia, through the efforts of the I. R.

Moravian-Silesian Economic Society, part of this project. This Society organised a network of meteorological stations (Fig. 9), collating the results of their observations (AS5). These are usually, however, relatively short and entries for many months are missing. Meteorological observations from Jihlava, where Andreas Sterly, a town councillor, kept daily observations in the 1816–1840 (1844) period, are an exemplary exception among these stations (Brázdil et al., 2007a). This

overview demonstrates the high importance of Hausner's observations, bridging a spatial gap on the one hand and covering very long period with otherwise only few observations for Moravia on the other.

### 5.3 Hausner's observations and climate fluctuations

Based on comparison of 1803–1830 with the reference period of 1961–1990 from the long temperature and precipitation series for Brno, the period of Hausner's observations was notably cooler in the winter half-year (October–March) and in June than the reference; of the remaining months, only May and August were slightly warmer (Fig. 10a). With the exception of February and March, all months were also more variable in terms of standard deviation than the reference period

(Fig. 10c). Wetter patterns in the Hausner period prevailed particularly in August, when precipitation totals characterised by variation coefficient were significantly less variable (Fig. 10b,d). In contrast, drier patterns prevailed mainly in February, from April to July and in November (Fig. 10b); precipitation totals from May to July and in December were also more variable (Fig. 10d).

25        Fluctuations in seasonal temperature and precipitation indices, appearing in Tables 1 and 2, may be compared with seasonal temperature means and precipitation totals from observations at the Brno station in the 1803–1830 period (data for Brno are taken from Brázdil et al., 2012a). Seasonal temperatures (Fig. 11) offer the closest accordance between Hausner's Buchlovice series and the Brno series (expressed by Pearson correlation coefficient $r$) for DJF ($r = 0.924$), followed by MAM

($r = 0.904$) and JJA ($r = 0.891$). Consistency becomes lower in SON ($r = 0.829$). However, all correlation coefficients are statistically significant at the 0.05 significance level. The correlation coefficient for annual series ($r = 0.912$) is also statistically significant and very high.

        Seasonal precipitation (Fig. 12), exhibits generally lower consistency between precipitation indices interpreted from Hausner's records for Buchlovice and measured totals in Brno. The JJA

patterns show the highest similarity ($r = 0.920$); lower correlations pertain to SON ($r = 0.806$), DJF ($r = 0.779$) and MAM ($r = 0.757$) patterns; however, all Pearson correlation coefficients are statistically significant at the 0.05 significance level. The correlation coefficient for annual precipitation series achieves the same value as that for SON.

        Despite generally close agreement between the Hausner series and those for Brno, some

instances of greater or smaller disagreement appear. This is particularly evident in interpretation of temperature/precipitation indices on a 7-degree ordinal scale (Pfister, 1992), an approach that cannot cover both positive and negative extremes well. Moreover, the interpretation of indices depends heavily on the comprehensiveness and degree of representation in Hausner's weather descriptions. These also depend on the intensity of weather manifestations, which is best expressed

by DJF, MAM and JJA temperatures and JJA precipitation (these are also expressed in the highest correlation coefficients appearing for these seasons and, with the exception of JJA, in higher values for temperatures compared with precipitation). While the distance between Brno and Buchlovice plays a generally negligible role in temperature patterns, the high spatial variability often associated with precipitation totals contributes to higher differences between the two places.

50        Fig. 13 compares annual variations of selected climatological characteristics at Buchlovice, as interpreted from Hausner's records for 1803–1830, with those from meteorological observations at the Brno, Buchlovice and Staré Město stations in 1961–1990. The annual number of precipitation



days according to Hausner's observations and to measurements at the Buchlovice rain-gauge station
(Fig. 13a) is nearly the same (120.1 and 117.9 days respectively). Hausner recorded a higher
frequency of such days especially in March, the summer months and October, while from
November to February their frequencies were lower. The annual number of days with strong winds
in Buchlovice (Fig. 13b) is higher than at the Brno airport station (67.4 and 48.1 days respectively);
the same holds for the monthly figures (except February), with a maximum in April in both series.
The higher numbers of such days at Buchlovice may clearly be attributed to the qualitative
evaluation of wind force by Hausner contrasting with the strictly-selected wind-speed thresholds for
Brno. The slightly higher annual numbers of days with thunderstorms (Fig. 13c) follow from
Hausner's data compared with the Staré Město station (14.9 and 12.2 days respectively). Annual
variations are nearly identical in general features for both Buchlovice and Staré Město, with a
maximum in June. The number of days with fog (Fig. 13d), consist of a smoothed annual
distribution with decreasing monthly frequencies from January to May–June followed by an
increase towards December for Buchlovice. The Staré Město station shows a consistently higher
numbers of such days, particularly from September to November (42 days with fog annually
compared to 19.2 such days at Buchlovice). This may be attributed to the position of the Staré
Město station in the valley of the River Morava, a location favouring the frequent occurrence of fog,
as well as the use of a strict fog definition related to guidelines for meteorological observations. The
horizon was also more-or-less limited in the vicinity of Hausner's dwelling in Buchlovice. A
relatively simplified mode of expression was selected for comparison of cloudiness (Fig. 13e),
dividing such days into those with clear sky, overcast sky and the remainder (cloudy sky). The
comparison of Hausner's records at Buchlovice with the Staré Město station is complicated by the
annual absence of 31.8 days or any interpretable records of them. However, it appears that
interpretation of Hausner's entries compared to Staré Město overestimates the number of clear days
(76.7 to 46.8 days annually) and underestimates the number of cloudy days (117.5 and 172.3 days
respectively).

**6 Conclusions**

The following conclusions may be drawn from this analysis of the weather diary records kept by
Šimon Hausner from Buchlovice in south-eastern Moravia, Czech Republic, in the first third of the
19th century:
(i) The Hausner's weather diary is a valuable source of meteorological and weather-related
information from south-eastern Moravia in the 1803–1830 period. It enables the creation of a
representative series of selected characteristics of temperature, precipitation, cloudiness, wind, other
meteorological phenomena, and weather-related phenological events and agricultural work.
(ii) Interpretation of Hausner's weather observations also enables the creation of series of weighted
temperature and precipitation indices for south-eastern Moravia, which may then be used in the
overlap period with the early instrumental meteorological observations in the Czech Republic for a
calibration/verification exercise in temperature and precipitation reconstructions that combine
documentary and instrumental data.
(iii) Series of 7-degree weighted temperature and precipitation indices derived from Hausner's
weather diary describe highly representative climate fluctuations in south-eastern Moravia during
the 1803–1830 period. The cold period in 1812–1816 and the dry period of 1805–1811 are
particularly worthy of note.
(iv) Hausner's data have great meteorological, climatological and phenological validity and
significantly supplement the Czech database of historical-climatological data and extend knowledge
of climatic variability of the first third of the 19th century in central Europe.

**Data availability.** The original daily weather records of Šimon Hausner are available in Moravian
Land Archives, Brno, fund G 138, catalogue no. 851. Data used in graphs are available from the
corresponding author.



**Competing interests.** The authors declare that they have no conflict of interest.

**Acknowledgements.** R.B., L.D., L.Ř. and A.V. acknowledge financial support from the Czech
Science Foundation for project no. 17-10026S. M.B. and P.Z. were supported by the SustES –
5    Adaptation strategies for sustainable ecosystem services and food security under adverse
environmental conditions project, no. CZ.02.1.01/0.0/0.0/16_019/0000797. Tony Long (Svinošice)
helped work up the English.

**Archival sources**

10   [AS1] Moravský zemský archiv Brno, fond B 16 Děkanské matriky uherskohradišťského děkanátu,
Kniha 416 Uherské Hradiště 1804–1808.
[AS2] Moravský zemský archiv Brno, fond G 138 Rodinný archiv Berchtoldů (1202)1494–1945,
inv. č. 349 Spory – s osvětimanským farářem Šimonem Hausnerem, 1825.
[AS3] Moravský zemský archiv Brno, fond G 138 Rodinný archiv Berchtoldů (1202)1494–1945,
15   inv. č. 851 Denní sledování počasí buchlovským farářem Šimonem Hausnerem 1803–1831.
[AS4] Moravský zemský archiv Brno, fond F 45 Velkostatek Buchlov (1514)1548–1949, inv. č.
449 Živelní škody, 1797–1848.
[AS5] Moravský zemský archiv Brno, fond G 82 Hospodářská společnost Brno 1769–1937, sign.
IV/3e Meteorologická pozorování (cartons 204–206).

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

**Climate of the Past Discussions**
EGU

**Table 1.** Weighted 7-degree temperature indices reconstructed from the daily weather records kept by Šimon Hausner in Buchlovice, 1803–1830.

| Year | J | F | M | A | M | J | J | A | S | O | N | D | DJF | MAM | JJA | SON | Ann |
|---|---|---|---|---|---|---|---|---|---|---|---|---|---|---|---|---|---|
| 1803 | -1 | -1 | 0 | 1 | 0 | 0 | 0 | 0 | -2 | -1 | 0 | -1 | - | 1 | 0 | -3 | -5 |
| 1804 | 2 | 0 | -1 | 0 | 0 | 0 | 0 | -1 | 0 | 0 | -3 | -2 | 1 | -1 | -1 | -3 | -5 |
| 1805 | -1 | -1 | -2 | -2 | -1 | -1 | -1 | -2 | 0 | -3 | -2 | 0 | -4 | -5 | -4 | -5 | -16 |
| 1806 | 1 | 1 | 0 | -1 | -1 | -1 | -1 | 0 | 1 | -1 | 1 | 1 | 2 | -2 | -2 | 1 | 0 |
| 1807 | 1 | 1 | -2 | -1 | 1 | 0 | 1 | 3 | 1 | 1 | 1 | 1 | 3 | -2 | 4 | 3 | 8 |
| 1808 | 0 | -1 | -3 | -1 | 1 | 1 | 2 | 2 | 2 | 0 | -2 | -2 | 0 | -3 | 5 | 0 | -1 |
| 1809 | -1 | -1 | -3 | -2 | 1 | 0 | 0 | 2 | 0 | -1 | -1 | 1 | -4 | -4 | 2 | -2 | -5 |
| 1810 | -1 | 0 | 1 | 0 | 0 | -1 | 0 | 1 | 2 | 0 | 1 | 0 | 0 | 1 | 0 | 3 | 3 |
| 1811 | -1 | -1 | 0 | 0 | 3 | 3 | 3 | 2 | 0 | 1 | -1 | -2 | -2 | 3 | 8 | 0 | 7 |
| 1812 | -1 | 0 | 1 | -3 | 0 | 0 | -1 | -1 | -2 | 2 | -2 | -2 | -3 | -2 | -2 | -2 | -9 |
| 1813 | -2 | 0 | -2 | 0 | 0 | -2 | -1 | 0 | -1 | -1 | -1 | 1 | -4 | -2 | -3 | -3 | -9 |
| 1814 | 0 | -3 | 0 | 0 | -3 | -2 | 0 | 0 | -2 | 0 | 0 | 0 | -2 | -3 | -2 | -2 | -10 |
| 1815 | 0 | 0 | 1 | 0 | 0 | 0 | -2 | -1 | -1 | 0 | -1 | -2 | 0 | 1 | -3 | -2 | -6 |
| 1816 | 1 | 0 | -1 | 0 | 0 | -1 | -2 | -2 | -2 | -1 | -1 | -1 | -1 | -1 | -5 | -4 | -10 |
| 1817 | 1 | 1 | 0 | -3 | 0 | 2 | 0 | 0 | 1 | 0 | 0 | 0 | 1 | -3 | 2 | 1 | 2 |
| 1818 | 0 | 0 | 0 | 1 | 0 | 0 | 1 | 1 | 0 | 0 | -1 | -2 | 0 | 1 | 2 | -1 | 0 |
| 1819 | -1 | 1 | 1 | 2 | 0 | 1 | 1 | 0 | 1 | 0 | 0 | -1 | -2 | 3 | 2 | 1 | 5 |
| 1820 | -2 | 0 | -1 | 1 | 0 | -2 | -1 | 2 | 0 | 1 | 0 | -1 | -3 | 0 | -1 | 1 | -3 |
| 1821 | 1 | -2 | -1 | 2 | -1 | -3 | -2 | -1 | -1 | 0 | 1 | 2 | -2 | 0 | -6 | 0 | -5 |
| 1822 | 2 | 1 | 1 | 1 | 1 | 1 | 1 | 0 | 0 | 2 | 0 | -1 | 5 | 3 | 2 | 2 | 9 |
| 1823 | -3 | 0 | 0 | -1 | 0 | -1 | -1 | 1 | 1 | 0 | 0 | 0 | -4 | -1 | -1 | 1 | -4 |
| 1824 | 0 | 1 | 0 | 0 | -2 | -1 | -1 | 0 | 1 | 1 | 1 | 2 | 1 | -2 | -2 | 3 | 2 |
| 1825 | 2 | -1 | -1 | 0 | 0 | 0 | 0 | 0 | 0 | 0 | 1 | 2 | 3 | -1 | 0 | 1 | 3 |
| 1826 | -2 | -1 | 0 | -1 | -2 | 0 | 2 | 2 | 1 | 1 | 0 | 1 | -1 | -3 | 4 | 2 | 1 |
| 1827 | 0 | -2 | 0 | 1 | 2 | 2 | 2 | 1 | 1 | 1 | -1 | 0 | -1 | 3 | 5 | 1 | 7 |
| 1828 | 1 | -1 | 1 | 1 | 0 | 0 | 2 | -2 | 0 | -1 | 0 | 0 | 0 | 2 | 0 | -1 | 1 |
| 1829 | -1 | -2 | -2 | 0 | -1 | -1 | 1 | -1 | 1 | -1 | -3 | -3 | -3 | -3 | -1 | -3 | -13 |
| 1830 | -3 | -2 | 0 | 0 | 0 | 1 | 1 | 1 | -2 | -1 | 0 | 0 | -8 | 0 | 3 | -3 | -5 |





**Table 2.** Weighted 7-degree precipitation indices reconstructed from daily weather records kept by Šimon Hausner in Buchlovice, 1803–1830.

| Year | J | F | M | A | M | J | J | A | S | O | N | D | DJF | MAM | JJA | SON | Ann |
|------|---|---|---|---|---|---|---|---|---|---|---|---|-----|-----|-----|-----|-----|
| 1803 | 2 | 0 | 0 | 0 | 2 | 1 | 0 | 0 | 0 | 1 | 0 | 2 | - | 2 | 1 | 1 | 8 |
| 1804 | 1 | 0 | 0 | 3 | -1 | 1 | 1 | 0 | 0 | 0 | 1 | 0 | 3 | 2 | 2 | 1 | 6 |
| 1805 | -2 | 1 | 0 | 0 | 2 | 0 | -2 | 0 | 0 | 0 | -2 | 0 | -1 | 2 | -2 | -2 | -3 |
| 1806 | 1 | 1 | 1 | -1 | -2 | -3 | 0 | 2 | 2 | -2 | 0 | -1 | 2 | -2 | -1 | 0 | -2 |
| 1807 | 0 | 1 | -1 | -1 | -1 | -1 | 0 | -2 | -2 | -1 | 2 | 1 | 0 | -3 | -3 | -1 | -5 |
| 1808 | -1 | -1 | -3 | 0 | -2 | -2 | -3 | -2 | 2 | -1 | 0 | 1 | -1 | -5 | -7 | 1 | -12 |
| 1809 | 0 | 0 | -2 | 0 | -2 | -2 | -2 | 0 | 1 | 0 | 0 | 2 | 1 | -4 | -4 | 1 | -5 |
| 1810 | -3 | 0 | 0 | 0 | 0 | -3 | -1 | -1 | -1 | -2 | -1 | 1 | -1 | 0 | -5 | -4 | -11 |
| 1811 | 0 | -1 | 0 | 1 | 0 | -1 | -1 | -2 | -1 | -2 | -2 | 1 | 0 | 1 | -4 | -5 | -8 |
| 1812 | 0 | 0 | 3 | 0 | 1 | 0 | 1 | 0 | -1 | 1 | 2 | 0 | 1 | 4 | 1 | 2 | 7 |
| 1813 | -3 | -3 | 0 | 0 | 1 | 2 | 2 | 1 | 1 | 2 | 0 | 1 | -6 | 1 | 5 | 3 | 4 |
| 1814 | 1 | -1 | -1 | -1 | -1 | 3 | -1 | 1 | 1 | 0 | -1 | 2 | 1 | -3 | 3 | 0 | 2 |
| 1815 | 1 | -1 | 1 | 0 | -1 | 2 | 1 | 1 | -3 | -2 | -2 | -1 | 2 | 0 | 4 | -7 | -4 |
| 1816 | 0 | 0 | 0 | 0 | 2 | 1 | 2 | 0 | 0 | 0 | 0 | -1 | -1 | 2 | 3 | 0 | 4 |
| 1817 | -1 | 1 | 1 | 1 | 0 | 0 | 0 | 0 | -2 | 1 | -3 | 0 | -1 | 2 | 0 | -4 | -2 |
| 1818 | -2 | -1 | 0 | -1 | 1 | -1 | 1 | 0 | 0 | 0 | -1 | -2 | -3 | 0 | 0 | -1 | -6 |
| 1819 | -1 | 1 | 0 | 0 | 0 | 0 | 0 | 1 | -2 | 0 | 0 | 0 | -2 | 0 | 1 | -2 | -1 |
| 1820 | 1 | 0 | 0 | -1 | 2 | 1 | -1 | -1 | 0 | 0 | 0 | -2 | 1 | 1 | -1 | 0 | -1 |
| 1821 | -2 | -2 | 0 | 0 | 0 | 1 | 2 | 1 | 1 | -1 | 0 | -1 | -6 | 0 | 4 | 0 | -1 |
| 1822 | 1 | -3 | 1 | -1 | -1 | -1 | 0 | 0 | 0 | -3 | -1 | -2 | -3 | -1 | -1 | -4 | -10 |
| 1823 | -1 | 2 | -2 | -2 | 0 | 0 | 1 | 1 | -1 | 0 | -1 | 0 | -1 | -4 | 2 | -2 | -3 |
| 1824 | -1 | -1 | -1 | 1 | 1 | 1 | 1 | 0 | -1 | 1 | 1 | 0 | -2 | 1 | 2 | 1 | 2 |
| 1825 | 0 | -2 | -1 | -2 | -2 | 0 | -1 | -1 | -1 | 0 | 0 | -1 | -2 | -5 | -2 | -1 | -11 |
| 1826 | -1 | 0 | 0 | -1 | 0 | 0 | -1 | -2 | -2 | 1 | 1 | 0 | -2 | -1 | -3 | 0 | -5 |
| 1827 | 2 | 2 | 2 | 0 | 0 | 1 | -3 | 0 | -1 | 0 | 2 | 0 | 4 | 2 | -2 | 1 | 5 |
| 1828 | 1 | 1 | 0 | 0 | 1 | 1 | 0 | 0 | -1 | 0 | -1 | 0 | 2 | 1 | 1 | -2 | 2 |
| 1829 | 1 | -1 | 0 | 2 | 0 | 0 | 0 | 0 | 0 | 0 | -1 | 1 | 0 | 2 | 0 | -1 | 2 |
| 1830 | -1 | 1 | -1 | 2 | 0 | -1 | -1 | -1 | 2 | 0 | -2 | 0 | 1 | 1 | -3 | 0 | -2 |



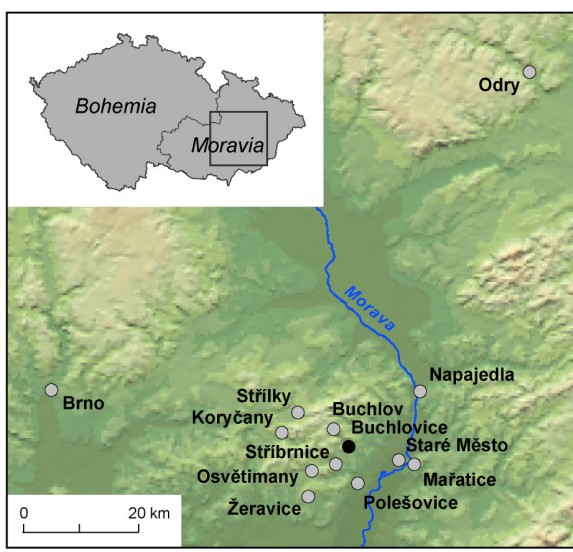

**Figure 1.** Locations of places within the Czech Republic referred to in this article.





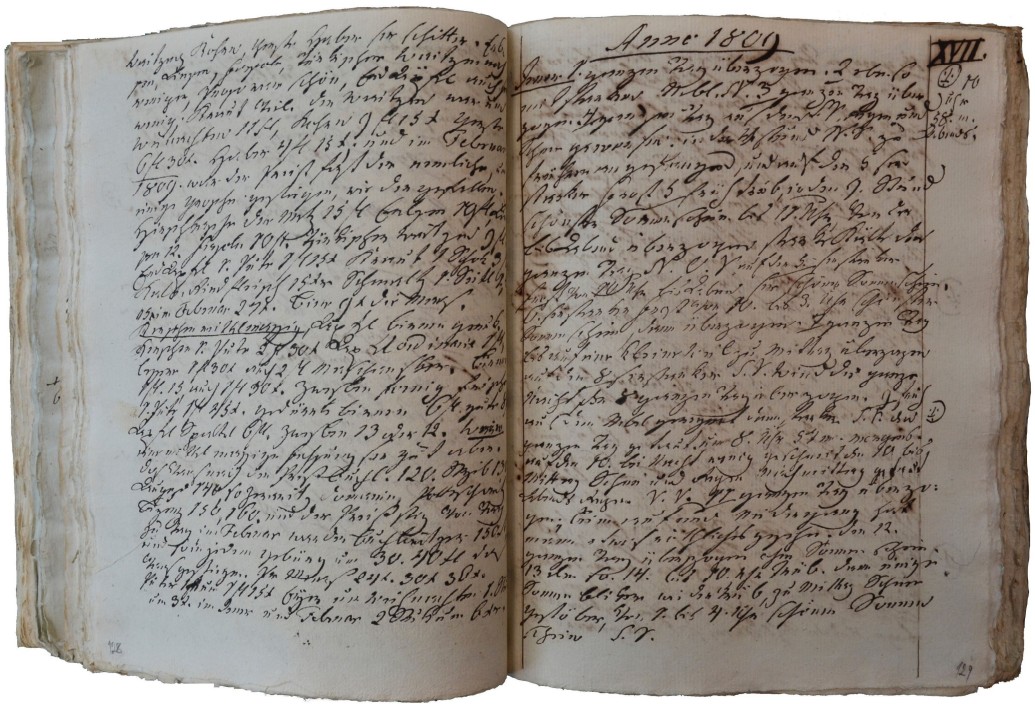

**Figure 2.** Example of pages from Hausner's weather diary, with records from the end of 1808 (left) and the beginning of 1809 (right) (AS3).





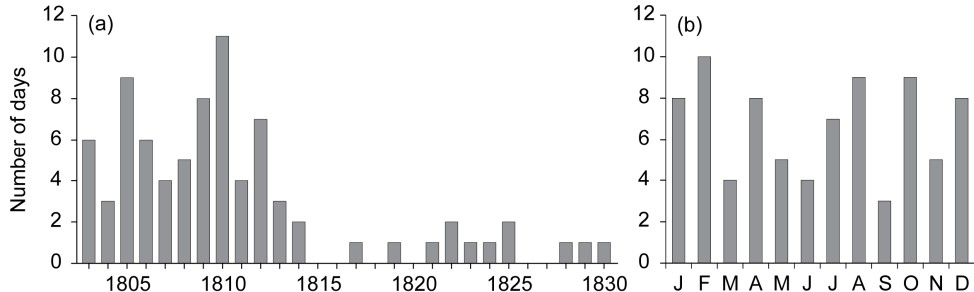

**Figure 3.** Days missing from daily weather records kept by Šimon Hausner at Buchlovice during 1803–1830 (a) and their annual variation (b).





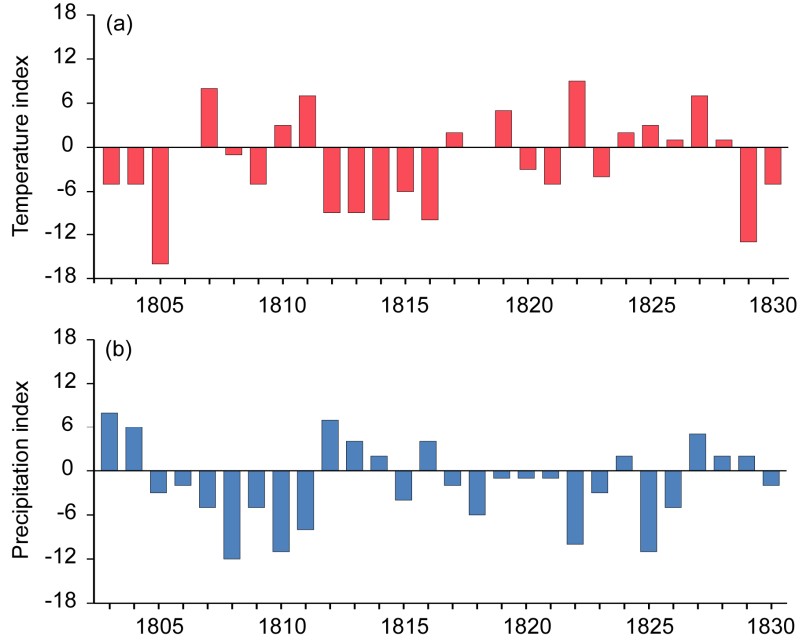

**Figure 4.** Fluctuations in annual temperature (a) and precipitation (b) indices at Buchlovice during 1803–1830, as derived from the weather diary kept by Šimon Hausner.





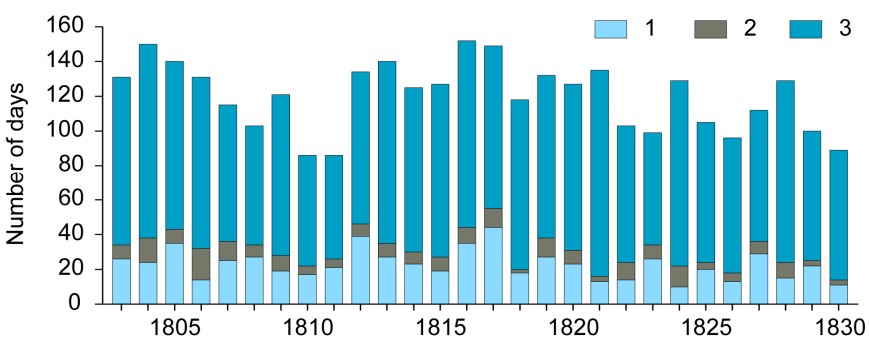

**Figure 5.** Fluctuations in the annual number of precipitation days (1 – solid, 2 – mixed, 3 – liquid) at Buchlovice during 1803–1830 derived from the weather diary kept by Šimon Hausner.





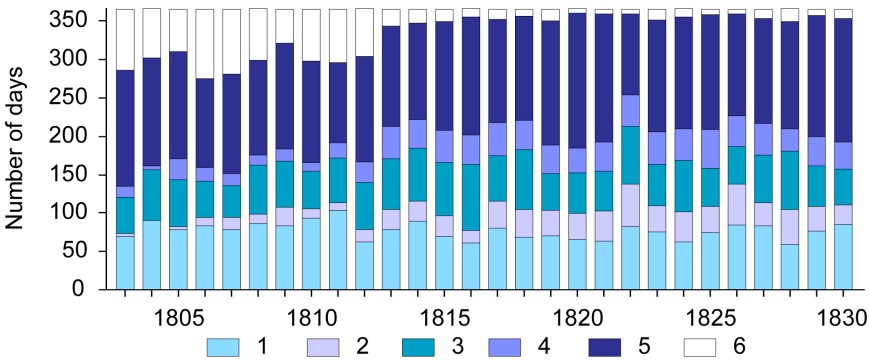

**Figure 6.** Fluctuations in annual cloudiness at Buchlovice during 1803–1830, derived from the weather diary kept by Šimon Hausner: 1) clear sky, 2) clear sky in one part of the day and half-covered sky in the other part, 3) half-covered sky, 4) half-covered sky in one part of the day and overcast in the other part, 5) overcast, 6) missing days or no information related to cloudiness.





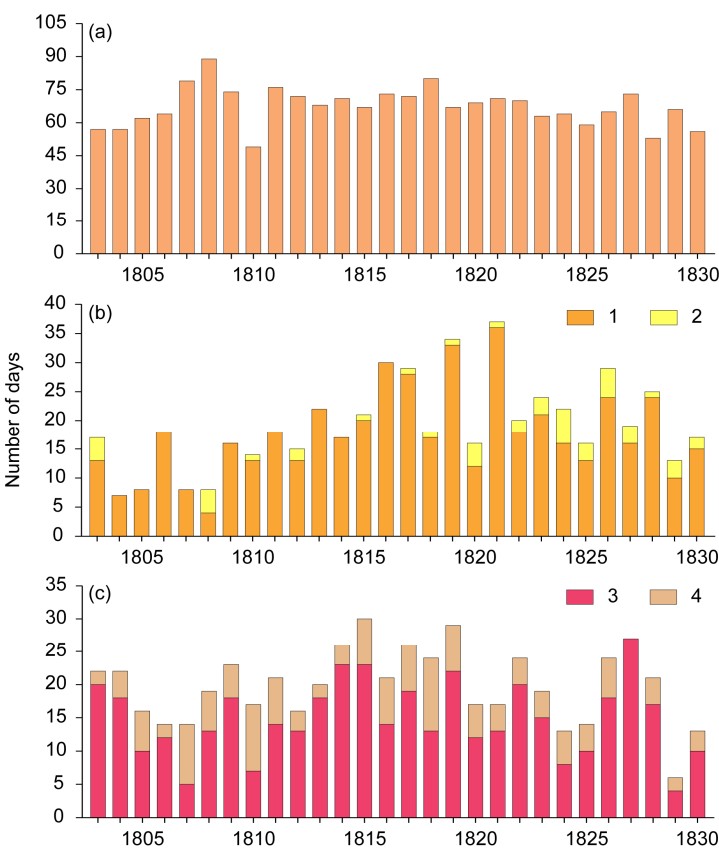

**Figure 7.** Fluctuations in the annual number of days with selected phenomena at Buchlovice during 1803–1830 derived from the weather diary kept by Šimon Hausner: (a) strong wind; (b) fog (1 – fog, 2 – foggy); (c) thunderstorm (3 – at Buchlovice, 4 – distant thunderstorm).





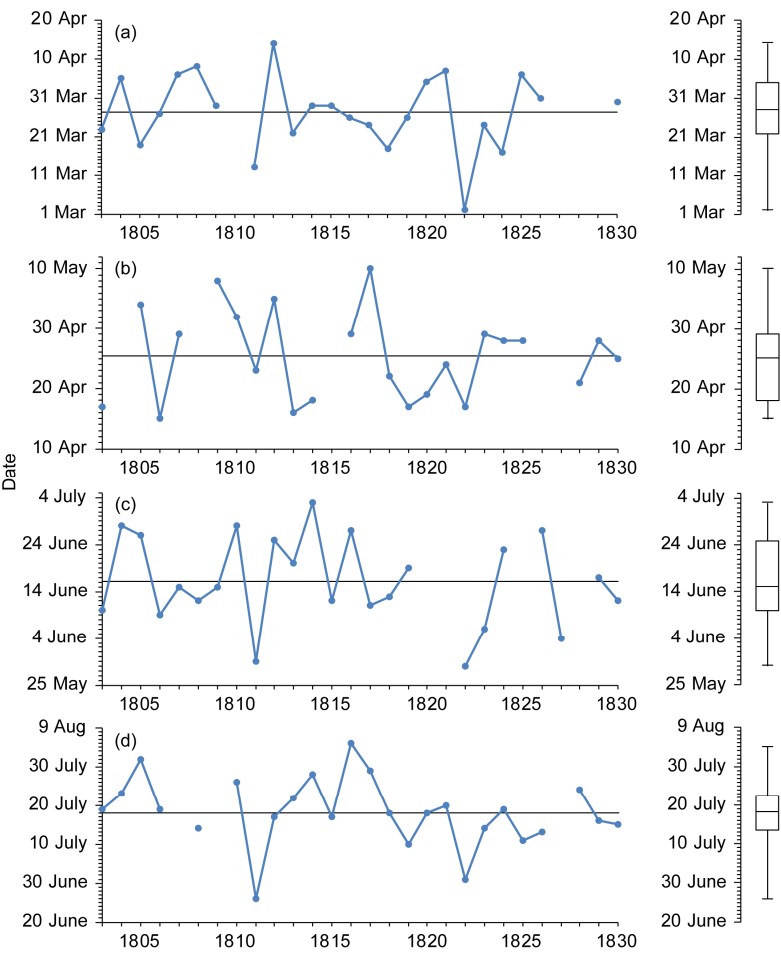

**Figure 8.** Fluctuations (left) and box-plots (right) of the beginnings of selected phenophases and agricultural work at Buchlovice during 1803–1830 derived from the weather diary kept by Šimon Hausner: (a) spring sowing of cereals; (b) blossoming of cherry trees; (c) blossoming of grapevine; (d) grain harvest.





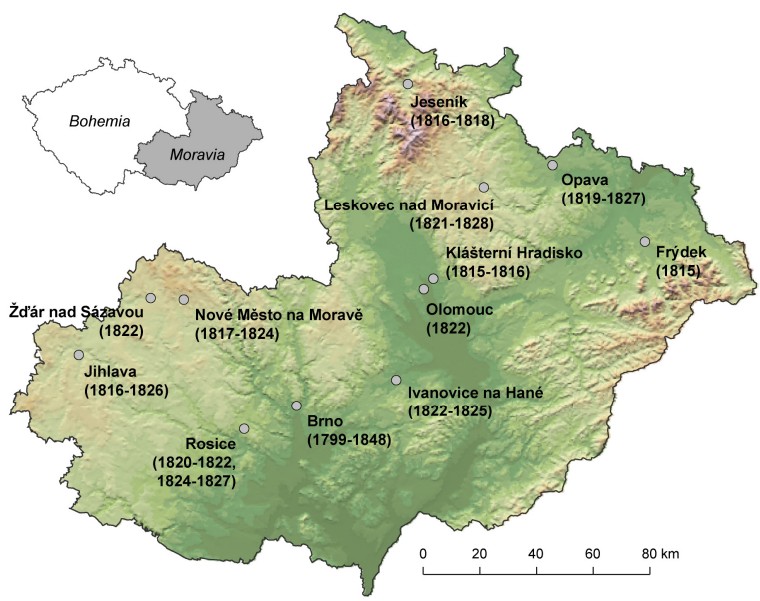

**Figure 9.** Network of meteorological stations in Moravia and Silesia as organised by the I. R. Moravian-Silesian Economic Society (years indicate available meteorological observations preserved in AS5).





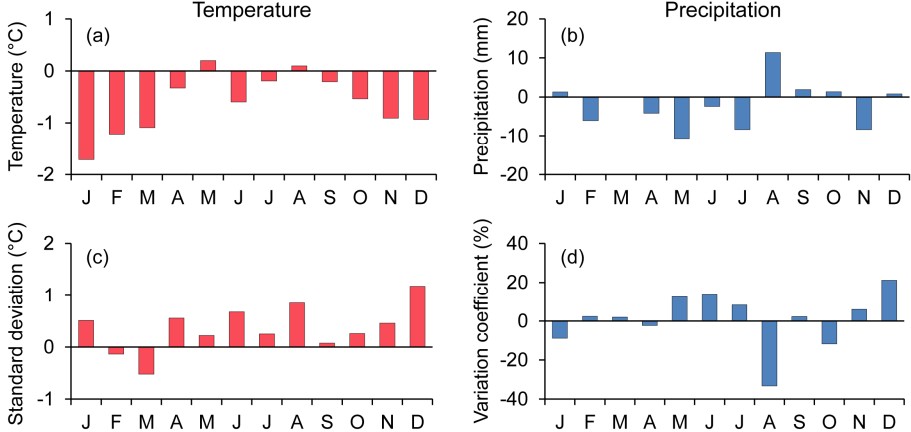

**Figure 10.** Comparison of monthly mean temperatures, precipitation totals and their variability between the periods 1803–1830 and 1961–1990 in Brno: differences in temperatures (a) and standard deviations (c), precipitation (b) and variation coefficients (d). Positive differences express higher values in 1803–1830 and negative differences in 1961–1990.





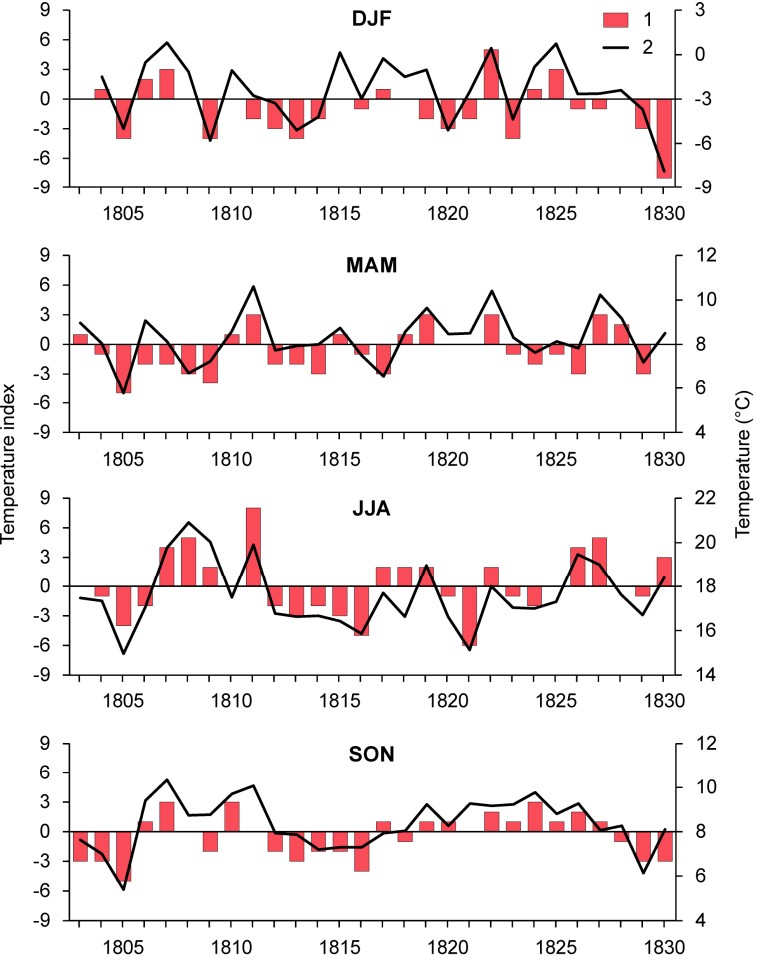

**Figure 11.** Fluctuations in weighted seasonal temperature indices in Buchlovice (1) and in mean seasonal temperatures in Brno (2) during the 1803–1830 period.

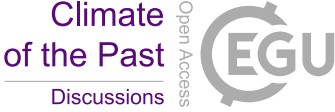

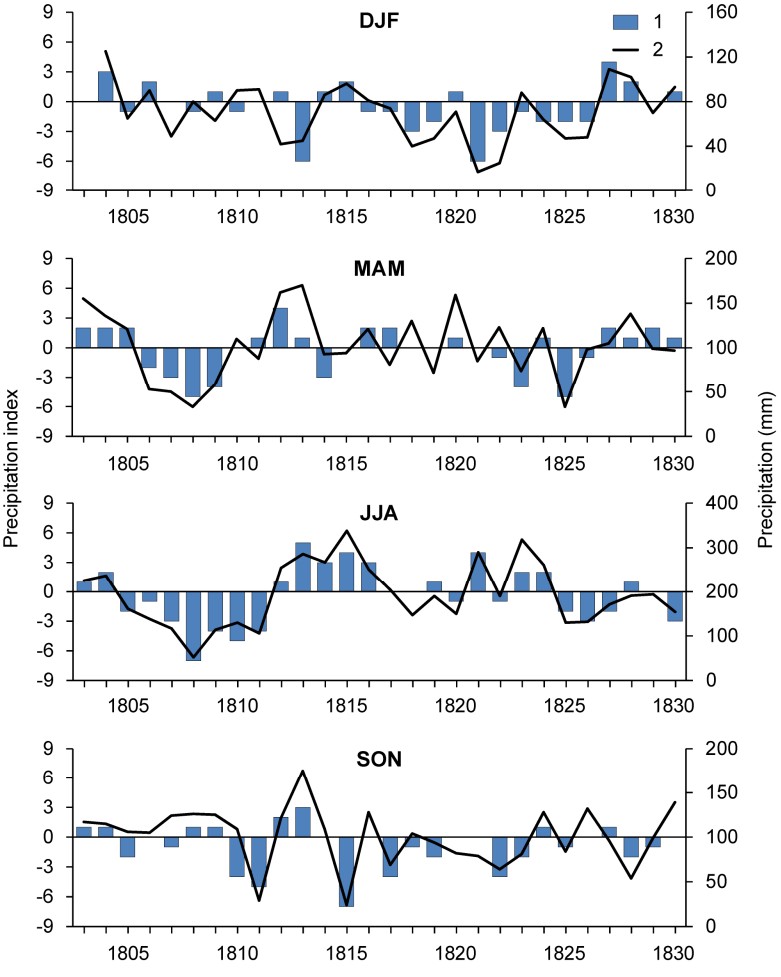

**Figure 12.** Fluctuations in weighted seasonal precipitation indices in Buchlovice (1) and in seasonal precipitation totals in Brno (2) during the 1803–1830 period.



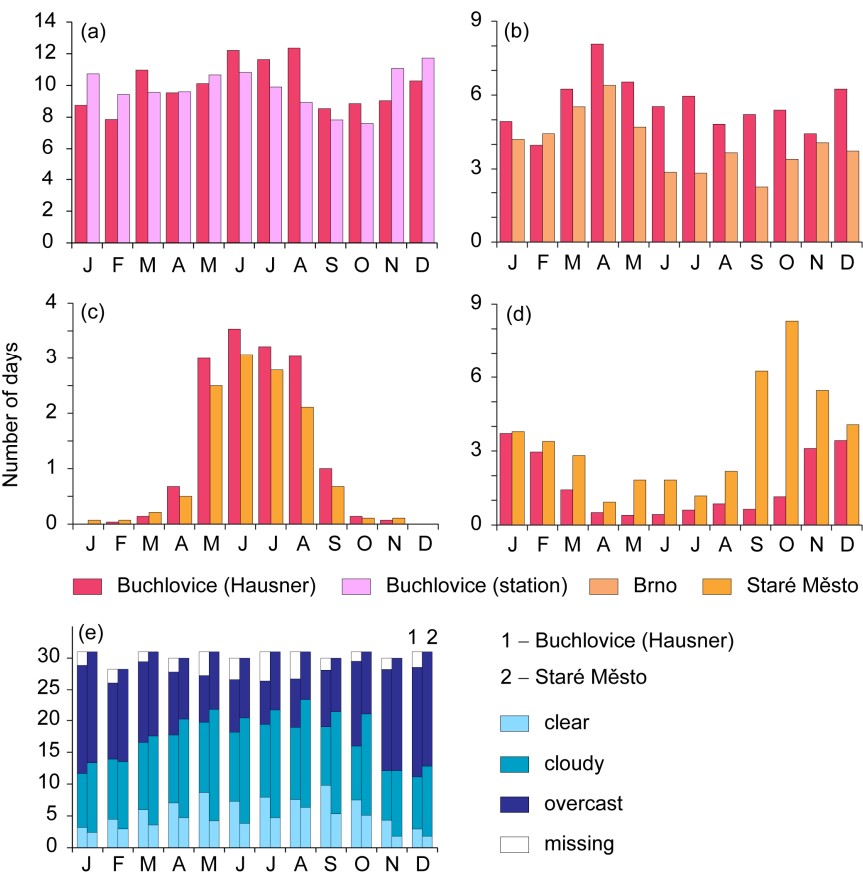

**Figure 13.** Comparison of annual variation of selected climatological characteristics at Buchlovice interpreted from Hausner's records for 1803–1830 and at three selected meteorological stations in 1961–1990: (a) number of precipitation days (with totals ≥0.1 mm for the Buchlovice rain-gauge station); (b) number of days with strong winds (with wind force ≥7ºB, i.e. wind speed ≥10.8 m s$^{-1}$ for Brno); (c) number of days with thunderstorm; (d) number of days with fog; (e) number of days with various degrees of cloudiness: clear days (0–2/10 of cloud cover at Staré Město), cloudy days (2.1–7.9/10 of cloud cover at Staré Město, sum of categories 2, 3 and 4 at Buchlovice – see Fig. 6) and overcast days (8–10/10 of cloud cover at Staré Město).