# Peer review of "The climate in south-east Moravia, Czech Republic, 1803–1830, based on daily weather records kept by the Reverend Šimon Hausner"

_Climate of the Past, 2019_

## Referee Comment (RC1) · Anonymous Referee #1 · 21 Apr 2019

The paper presents a very detailed analysis of weather and climate in south-east Moravia in the years 1803–1830 based on a newly-discovered daily weather diary from Buchlovice written by Šimon Hausner, a priest in Buchlovice. Although meteorological observations exist for this time period for some stations in the Czech Republic, including the closest station in Brno, the value of such long series of visual observations is very important, not only for improving the climate knowledge of the region, but even more for estimation of the usefulness of that kind of weather excerpts for climate reconstruction, including estimation of its uncertainty. The main weakness of the paper, which necessarily must be supplemented, is a lack of information concerning the way that air temperature and precipitation values are attributed to a specific index in the 7-

degree scale. In Section 3.2 there should be information about threshold values used in the process of indexation based on monthly frequencies of warm or cold days in case of temperature and number of days with precipitation in case of precipitation. Do you use data from Brno station for this purpose, e.g. number of days with precipitation? Why did you not make daily indexation using e.g. a 3-degree scale? Does Hausner's weather diary allow for such indexation or not? When he started weather observations, Hausner was a mature man, thus probably his weather descriptions concerning its extremity were related to his weather experience in the late 18th century, a period which was warmer than 1803–1830. This is probably the reason why your indexation revealed significantly more months described as extremely cold and very cold compared to extremely warm and very warm, in particular in winter months (Table 1). For the entire year the statistic is the following: for -3 and +3 ( 13 and 4, respectively), and for -2 and +2 (37 and 23). The second possibility is that the person who made the indexation compared Hausner's descriptions of weather with the present period, which is also warmer. My next doubt concerns the reference period: why did you not use the latest normal period 1981–2010, as recommended by the WMO? Such comparison will give a better estimate of climate change and variability between historical and present periods.

Minor remarks: 1. Page 1, last line – I suggest to add here for the 18th century the recently published paper by Filipiak et al. (2019) presenting results for Gdansk for the period 1721–1786 based on Reyger's weather observations (https://doi.org/10.1002/joc.5845), 2. Quite a lot of shortcomings which exist in the paper should be supplemented, e.g. p. 10, lines 18-19: "With the exception of February and March, all months were also more variable in terms of standard deviation than the reference period", p. 10, lines 39-40: "Despite generally close agreement between the Hausner series (ref. comm.: there are a lot of variables analysed in the paper: does the statement concern all variables or only temperature and precipitation?) and those for Brno, some instances of greater or smaller disagreement appear", etc. 3. Fig. 13 – I suggest more contrastive colours be used to show data from Brno and Stare Mesto.

It is difficult to guess which of the mentioned stations the data in Fig. c represents, 4. Fig. 13 – in the caption there is information that strong winds were estimated as those with force ïĆş 7oB. In Section 3.1 there is no information on how this was estimated based on Hausner's weather descriptions. I suggest this information be added, 5. The authors should maybe reconsider the presentation of Section 3.1 (or part of it) in the form of a Table, in particular for temperature and precipitation. It seems to me that the text will then be more clear for readers. I can recommend acceptance of the paper for publication in the Climate of the Past journal only on the condition that the listed remarks and suggestions will be satisfactorily taken into account.

---

## Referee Comment (RC2) · Anonymous Referee #2 · 20 May 2019

This is an interesting paper that provides a new meteorological record for a number of parameters for a region and period where none currently exists. It follows a relatively standard methodology and provides some interesting findings that can be built on in the future to create a more systematic climatological record for the region. I'm happy for it to be published, although I would like the following three points to be addressed:

1. The significance of the missing datapoints is not clearly addressed, and figures 5-7 and 13 erroneously use exact numbers of days, despite the missing data. I would like to see a discussion of the distribution of the missing days and - unless the distribution of missing days makes this invalid - the exact days in these figures to be translated

into percentages 2. There is nothing to show to the reader how the 7-point index figures were arrived at. I'd like to see some kind of calibration table, and/or examples of years under each classification, and/or a comparison of classifications assigned between the different researchers and a discussion of how the final classification was arrived at (assuming this was done) 3. It would be good if the authors could provide possible climatological reasons for the difference in values between the reconstruction and contemporary period.

Detailed comments are below: Section 3.2. Can you give details of the distribution of missing days? Also this isn't really statistical analysis, instead it's a discussion of the methodology used to generate weather indices, so the title should be changed. I'd also like more detail. What descriptors would produce a rank of -2, as opposed to -3, for instance? Some kind of table that details terms or conditions related to each category are necessary, and/or examples of years falling under each category.

Page 6 lines 5-6-9. To what extent is this a relic of the reconstruction methodology, rather than a 'true' representation of the variability? This seems to be the same for the wetness indicators (lines 33-35)

Sections 4.1.2-4 and Figures 5-7. I'm not sure about the suitability of using absolute counts of days, given that the number of recorded days is so variable. Perhaps percentages would be better? This will depend of course on the distribution of missing days – if they are relatively regularly spaced through the year this would give a stronger justification for using percentages.

Section 5.2 and figure 9. I think this is in the wrong place in the argument, and should come in the opening sections (before the methodology). This is describing the context of the Hausner diaries, not anything within the diaries themselves.

Page 10 lines 15-24. Can you give any climatological suggestion for the substantially larger temperature increase to the modern period in winter compared to summer? Also, why do we see such a variation in both quantity and stdev of precipitation in August?

Can this be explained by one or two years with particularly heavy rainfall during this month?

Page 10 lines 47-49. Needs a reference

Page 11 – up to line 26. These comparisons are invalid, given the number of missing days in Hausner's records. Again, the use of percentages would be better, if this can be justified due to the distribution of the missing days. Also – as with the section above – can you suggest any climatological reason for the variation observed? It is true that location of the meteorological stations is likely to have an impact, but you are also looking at datasets separated by 130 years of a changing climate.

---

## Author Comment (AC1) · 24 May 2019

The paper presents a very detailed analysis of weather and climate in south-eastMoravia in the years 1803–1830 based on a newly-discovered daily weather diary from Buchlovice written by Šimon Hausner, a priest in Buchlovice. Although meteorological observations exist for this time period for some stations in the Czech Republic, including the closest station in Brno, the value of such long series of visual observations is very important, not only for improving the climate knowledge of the region, but even more for estimation of the usefulness of that kind of weather excerpts for climate reconstruction, including estimation of its uncertainty. RESPONSE: We would like to thank the reviewer for many very important comments to which we are trying respond below.

The main weakness of the paper, which necessarily must be supplemented, is a lack of information concerning the way that air temperature and precipitation values are attributed to a specific index in the 7-degree scale. In Section 3.2 there should be information about threshold values used in the process of indexation based on monthly frequencies of warm or cold days in case of temperature and number of days with precipitation in case of precipitation. Do you use data from Brno station for this purpose, e.g. number of days with precipitation? RESPONSE: The methodology of interpretation of temperature and precipitation indices was described in Section 3.2 in more detail as follows: "With respect to the character of Hausner's daily weather records, it became impossible apply some new quantitative approaches to interpret monthly temperature and precipitation indices (see e.g. Fernández-Fernández et al., 2017; Filipiak et al, 2019). From this reason we used a broadly applied approach of Pfister (1992), combining different kinds of sources and their expert evaluation. Information related to temperature patterns was used to interpret monthly temperature indices by expression on a 7-degree scale: –3 extremely cold, –2 very cold, –1 cold, 0 normal, 1 warm, 2 very warm, 3 extremely warm (Pfister, 1992). Interpretation of temperature indices took into account the broad scale of indicators derived from Hausner's records: the monthly frequencies of cold days (severe frost, frost, cold, very cold) and warm days (warm, very warm, hot, very hot, mild), warm and cold winds, monthly summary reports, early and late beginnings of certain phenophases and agricultural work and also, to some extent, cloudiness (e.g. clear and overcast days) and precipitation (state of precipitation, monthly temperature–precipitation relationships). Own interpretation was realised in

[Figure]

following iterations: (i) Pfister (1992) recommended attribution of regularly distributed 7-degre indices to dataset ordered from the lowest to the highest values: index –3 he used for 8.3% lowest values, –2 for 16.6% following values, further with always 16.6% of values for each following indices (–1, 0, 1, 2) up to 8.3% of highest values attributed to index 3. This approach was applied separately to the monthly frequencies of cold days and warm days in 1803–1830, which allowed attribute any index to each of months, when in indexing of the month of winter half-year rather cold days and in the months of summer half-year rather warm days were preferred. (ii) Monthly indices from point (i) were further evaluated with respect to the structure of cold and warm days looking on their intensity, based on which the corresponding month could be moved to the neighbour category. For example, higher portion of hot or very hot days was a reason for moving to "warmer" category (e.g. from index 1 to index 2), similarly as higher portion of weak frosts or cold days compare to severe frosts and very cold days (e.g. from index –2 to index –1), and opposite. Also indication of warm and cold winds as expression of character of air advection was considered. (iii) All indices from iteration (ii) were further considered with respect to monthly temperature summaries and earlier/later onset of phenophases, indicating cooler or warmer patterns of preceding months. As additional parameter also information about cloudiness was used (days with higher sunshine duration are warmer compared to cloudy days). Also occurrence of snowfall or snow cover indicated cooler patterns. For months of the summer half-year also relationship warm/dry and cold/wet month was considered. Described facts could again cause moving of some months to another neighbour category. Corresponding indices, fixed after the third iteration, were then used as a final version of weighted monthly temperature indices. Also precipitation indices were interpreted in similar fashion: –3 extremely dry, –2 very dry, –1 dry, 0 normal, 1 wet, 2 very wet, 3 extremely wet (Pfister, 1992). The interpretation of monthly precipitation indices was based again on several indicators Hausner's records: monthly frequencies of precipitation days, with particular reference to type of precipitation (e.g. snow, drizzle, rain, snow with rain), to precipitation intensity and duration of precipitation spells

(as specified in daily records) and to summary monthly reports as well as other indications of wet or dry patterns (e.g. effects on agricultural crops or work in the fields). The own interpretation included three iterations: (i) The numbers of monthly precipitation days of the given months in 1803–1830 (28 years) was ordered from the lowest to the highest numbers. Following the above percentage distribution by Pfister (1992), corresponding 7-degree indices between –3 and 3 were formally added to individual months. (ii) Based on additionally reported type, intensity and duration of precipitation, the corresponding months remained in preliminary defined degree from point (i) or it was moved to any neighbour degree. For example, more days with drizzle, short or small precipitation were favourable to adding "more dry" index (e.g. from index 2 to index 1) or days with the whole-day rain or heavy precipitation during thunderstorms could be favourable for opposite attribution of "more wet" index (e.g. from index –1 to index 0). (iii) All indices from point (ii) were further evaluated from point of view of summary information of precipitation character of the months or any other precipitation-sensitive information. Similar as in the previous iteration (ii), some months could be moved to other degree. After these three iterations corresponding monthly precipitation indices were considered as final." It means, that we combined quantitative approach (based on distribution of corresponding data and number of days with some characteristics) with an expert approach based on expert knowledge of climate in the Czech Lands. There is clear that final version of monthly indices is influenced by character of basic data, their interpretation and climatic experience of the researcher. But having no measured data we cannot expect more. Some tests of a quality of indices interpreted can be the high and statistically significant correlation coefficients of our indices with measured temperatures/precipitation series in Brno (Section 5.3). Based on it we believe that our rather subjective interpretation express quite well corresponding temperature and precipitation variability (fluctuations) following from the meteorological measurements of the Brno station. Replaying to your question on the instrumental data of the Brno station, we did not use them in the interpretation of indices from Hausner's data, i.e. temperature and precipitation measurements in Brno are independent. Brno series

were used only for comparison with Hausner's indices in Section 5.3.

Why did you not make daily indexation using e.g. a 3-degree scale? Does Hausner's weather diary allow for such indexation or not? RESPONSE: We are really sorry, but Hausner's records did not allow us to do any systematic daily indexation as used, for example, in the paper by Filipiak et al. (2019). From this reason we could not use also an indexation approach, which was applied in this important reported paper.

When he started weather observations, Hausner was a mature man, thus probably his weather descriptions concerning its extremity were related to his weather experience in the late 18th century, a period which was warmer than 1803–1830. This is probably the reason why your indexation revealed significantly more months described as extremely cold and very cold compared to extremely warm and very warm, in particular in winter months (Table 1). For the entire year the statistic is the following: for -3 and +3 (13 and 4, respectively), and for -2 and +2 (37 and 23). RESPONSE: Each description of weather, which is not based on instrumental measurements, has a subjective feature and is influenced by many factors, including the observer's experience. We may only speculate how much he was influenced by the experience from the previous period. But why we should primarily expect, that the number of corresponding negative and positive extremes should be more or less the same, i.e. to be in agreement with any "normal" patterns? It would be more or less same, when we would stop creation of corresponding indices after the first iteration (see Section 3.2). But from Fig. 10 clearly follows, that the 1803-1830 period, belonging to the last part of the LIA, was much cooler compared to more recent time, influenced by a global warming. In our opinion, it implies that the number of negative extremes should be higher than the number of positive extremes. It is also confirmed, for example, by the fact that Büntgen et al. (2015) spoken of the 1810s as the coolest decade in central Europe in the past three centuries. Cole-Dai et al. (2009) referred even to this time as probably the coldest decade in the last 500 years or more in the Northern Hemisphere.

The second possibility is that the person who made the indexation compared Hausner's

descriptions of weather with the present period, which is also warmer. RESPONSE: We do not thing that it was particularly this case. On the other hand, each kind of interpretation of "descriptive" daily weather records bears some subjective features. From this reason interpretations of indices based on Hausner's records were compared with measured temperatures/precipitation from Brno (see Section 5.3) which showed quite a good agreement in character of fluctuations in both types of series.

My next doubt concerns the reference period: why did you not use the latest normal period 1981–2010, as recommended by the WMO? Such comparison will give a better estimate of climate change and variability between historical and present periods. RESPONSE: The reason for the use of the 1961–1990 reference period are following: (i) in the quoted WMO material, the period from 1961 to 1990 has been retained as a standard reference period for long-term climate change assessments; (ii) majority of climatological papers up to now is using the standard reference period 1961–1990, i.e. from point of view of possible comparisons it is better to preserve this period; (iii) the 1961–1990 period is not so strongly influenced by recent global warming as the 1981–2010 period; (iv) according to Czech representative in WMO Dr. R. Tolasz, who participated in preparation of this material, "30-year period out of standard normals should be used rather for evaluation of more recent deviations (e.g. on the monthly level)".

Minor remarks: 1. Page 1, last line – I suggest to add here for the 18th century the recently published paper by Filipiak et al.(2019) presenting results for Gdansk for the period 1721–1786 based on Reyger's weather observations (https://doi.org/10.1002/joc.5845), RESPONSE: Accepted and included into the manuscript.

2. Quite a lot of shortcomings which exist in the paper should be supplemented, e.g. p. 10, lines 18-19: "With the exception of February and March, all months were also more variable in terms of standard deviation than the reference period", RESPONSE: Accepted and corrected as: "With the exception of February and March, all months

were also more variable according to standard deviation than in the reference period."

p. 10, lines 39-40: "Despite generally close agreement between the Hausner series (ref. comm.: there are a lot of variables analysed in the paper: does the statement concern all variables or only temperature and precipitation?) and those for Brno, some instances of greater or smaller disagreement appear", etc. RESPONSE: Accepted and corrected as: "Despite generally close agreement between the Hausner temperature and precipitation series and those for Brno, some instances of greater or smaller disagreement appear." We are sorry, but other shortcomings in the manuscript are not specified, i.e. there is difficult to respond. The manuscript was corrected for English style by a native speaker.

3. Fig. 13 – I suggest more contrastive colours be used to show data from Brno and Stare Mesto. It is difficult to guess which of the mentioned stations the data in Fig. c represents, RESPONSE: Accepted and corrected.

4. Fig. 13 – in the caption there is information that strong winds were estimated as those with force ï ÌĄC Ìğs 7oB. In Section 3.1 there is no information on how this was estimated based on Hausner's weather descriptions. I suggest this information be added, RESPONSE: To specify the interpretation of days with strong winds according to Hausner's observations, following sentence was added as the second on the beginning of Section 4.1.4: "As days with strong winds were interpreted those in which Hausner mentioned strong or very strong wind, very windy weather, "awful" wind, extraordinary wind, windstorm or blizzard (see Section 3.1, point (iv))."

5. The authors should maybe reconsider the presentation of Section 3.1 (or part of it) in the form of a Table, in particular for temperature and precipitation. It seems to me that the text will then be more clear for readers. RESPONSE: We have some doubts, that creation of the table would be more simple for readers than presentation of terminology used in descriptive form because the table should be relatively large and complicated. Moreover, it would be inconsistent to do a table only for temperature and precipitation

terminology. From this reason we would be more happy to let it in the recent form. We believe that reader interested in topic will be able to go via this well-structured text without any troubles.

I can recommend acceptance of the paper for publication in the Climate of the Past journal only on the condition that the listed remarks and suggestions will be satisfactorily taken into account. RESPONSE: We tried to respond to every of your important comments in the best possible way with a hope that responses could be taken by you as satisfactorily.

---

## Author Comment (AC2) · 24 May 2019

This is an interesting paper that provides a new meteorological record for a number of parameters for a region and period where none currently exists. It follows a relatively standard methodology and provides some interesting findings that can be built on in

the future to create a more systematic climatological record for the region. I'm happy for it to be published, although I would like the following three points to be addressed: RESPONSE: We would like to thank the reviewer for many very important comments to which we are trying respond below.

1. The significance of the missing datapoints is not clearly addressed, and figures 5-7 and 13 erroneously use exact numbers of days, despite the missing data. I would like to see a discussion of the distribution of the missing days and - unless the distribution of missing days makes this invalid - the exact days in these figures to be translated into percentages RESPONSE: Accepted, see our response to Section 3.2 of your comments. Moreover, the numbers of missing dates were included directly into Figs. 5 and 7. In Fig. 6 is already included category 6 - missing days or no information related to cloudiness. As for Fig. 13, please see our detail response to page 11 of your comments.

2. There is nothing to show to the reader how the 7-point index figures were arrived at. I'd like to see some kind of calibration table, and/or examples of years under each classification, and/or a comparison of classifications assigned between the different researchers and a discussion of how the final classification was arrived at (assuming this was done) RESPONSE: Accepted, the methodology of creation of 7-point indices was complemented in Section 3.2 as follows: "With respect to the character of Hausner's daily weather records, it became impossible apply some new quantitative approaches to interpret monthly temperature and precipitation indices (see e.g. Fernández-Fernández et al., 2017; Filipiak et al, 2019). From this reason we used a broadly applied approach of Pfister (1992), combining different kinds of sources and their expert evaluation. Information related to temperature patterns was used to interpret monthly temperature indices by expression on a 7-degree scale: –3 extremely cold, –2 very cold, –1 cold, 0 normal, 1 warm, 2 very warm, 3 extremely warm (Pfister, 1992). Interpretation of temperature indices took into account the broad scale of indicators derived from Hausner's records: the monthly frequencies of cold days (severe frost, frost, cold, very cold) and warm days (warm, very warm, hot, very hot, mild), warm and cold winds, monthly summary reports, early and late beginnings of certain phenophases and agricultural work and also, to some extent, cloudiness (e.g. clear and overcast days) and precipitation (state of precipitation, monthly temperature–precipitation relationships). Own interpretation was realised in following iterations: (i) Pfister (1992) recommended attribution of regularly distributed 7-degre indices to dataset ordered from the lowest to the highest values: index –3 he used for 8.3% lowest values, –2 for 16.6% following values, further with always 16.6% of values for each following indices (–1, 0, 1, 2) up to 8.3% of highest values attributed to index 3. This approach was applied separately to the monthly frequencies of cold days and warm days in 1803–1830, which allowed attribute any index to each of months, when in indexing of the month of winter half-year rather cold days and in the months of summer half-year rather warm days were preferred. (ii) Monthly indices from point (i) were further evaluated with respect to the structure of cold and warm days looking on their intensity, based on which the corresponding month could be moved to the neighbour category. For example, higher portion of hot or very hot days was a reason for moving to "warmer" category (e.g. from index 1 to index 2), similarly as higher portion of weak frosts or cold days compare to severe frosts and very cold days (e.g. from index –2 to index –1), and opposite. Also indication of warm and cold winds as expression of character of air advection was considered. (iii) All indices from iteration (ii) were further considered with respect to monthly temperature summaries and earlier/later onset of phenophases, indicating cooler or warmer patterns of preceding months. As additional parameter also information about cloudiness was used (days with higher sunshine duration are warmer compared to cloudy days). Also occurrence of snowfall or snow cover indicated cooler patterns. For months of the summer half-year also relationship warm/dry and cold/wet month was considered. Described facts could again cause moving of some months to another neighbour category. Corresponding indices, fixed after the third iteration, were then used as a final version of weighted monthly temperature indices. Also precipitation indices were interpreted in similar fashion: –3 extremely dry, –2 very dry, –1

dry, 0 normal, 1 wet, 2 very wet, 3 extremely wet (Pfister, 1992). The interpretation of monthly precipitation indices was based again on several indicators Hausner's records: monthly frequencies of precipitation days, with particular reference to type of precipitation (e.g. snow, drizzle, rain, snow with rain), to precipitation intensity and duration of precipitation spells (as specified in daily records) and to summary monthly reports as well as other indications of wet or dry patterns (e.g. effects on agricultural crops or work in the fields). The own interpretation included three iterations: (i) The numbers of monthly precipitation days of the given months in 1803–1830 (28 years) was ordered from the lowest to the highest numbers. Following the above percentage distribution by Pfister (1992), corresponding 7-degree indices between –3 and 3 were formally added to individual months. (ii) Based on additionally reported type, intensity and duration of precipitation, the corresponding months remained in preliminary defined degree from point (i) or it was moved to any neighbour degree. For example, more days with drizzle, short or small precipitation were favourable to adding "more dry" index (e.g. from index 2 to index 1) or days with the whole-day rain or heavy precipitation during thunderstorms could be favourable for opposite attribution of "more wet" index (e.g. from index –1 to index 0). (iii) All indices from point (ii) were further evaluated from point of view of summary information of precipitation character of the months or any other precipitation-sensitive information. Similar as in the previous iteration (ii), some months could be moved to other degree. After these three iterations corresponding monthly precipitation indices were considered as final." It means, that we combined quantitative approach (based on distribution of corresponding data and number of days with some characteristics) with an expert approach based on expert knowledge of climate in the Czech Lands. There is clear that final version of monthly indices is influenced by character of basic data, their interpretation and climatic experience of the researcher. But having no measured data we cannot expect more. Some tests of a quality of indices interpreted can be the high and statistically significant correlations of our indices with measured temperatures/precipitation series in Brno (Section 5.3). Based on it we believe that our rather subjective interpretation express quite well corresponding temperature and precipitation variability following from the meteorological measurements of the Brno station.

3. It would be good if the authors could provide possible climatological reasons for the difference in values between the reconstruction and contemporary period. RESPONSE: Accepted. See partly our responses to page 10 of your comments below. Comparison of the reconstruction and contemporary period was motivated by the fact to put Hausner's period into context of long-term climate variability. We believe that for understanding of this context Fig. 10 and the first paragraph of Section 5.1 are sufficient. While the 1961–1990 period clearly reflects the first part of intense temperature increase related to recent global warming (caused by increase in GHG gases), the cooler reconstruction period is located in the time of effects of natural climate variability (e.g. volcanic activity) during the last phase of LIA. For example, Büntgen et al. (2015) spoken of the 1810s as the coolest decade in central Europe in the past three centuries. Cole-Dai et al. (2009) referred to this time as probably the coldest decade in the last 500 years or more in the Northern Hemisphere. These papers confirm results of our comparison. To do any real sophisticated climatological analysis of differences in both periods would need to take into account many climatic triggers (solar and volcanic activity, anthropogenic factor) and large scale climate drivers (e.g. different circulation indices like NAOI, SOI, PDO, etc.), But such analysis is going clearly out of the scope of this paper and would need a separate special research.

Detailed comments are below: Section 3.2. Can you give details of the distribution of missing days? Also this isn't really statistical analysis, instead it's a discussion of the methodology used to generate weather indices, so the title should be changed. I'd also like more detail. What descriptors would produce a rank of -2, as opposed to -3, for instance? Some kind of table that details terms or conditions related to each category are necessary, and/or examples of years falling under each category. RESPONSE: Accepted. Concerning of the distribution of missing days, it is clearly presented in Fig. 3 and for individual years described in the first paragraph of Section 3.2, where we add

also relative expression (the maximum of 11 days in 1810 represents only 3% of days and for other 19 years the number of missing days is less than 1% or even zero). To fulfill your request, we extended corresponding description as follows: "A total of 80 days is missing (i.e. around 3 days per year), tending towards the years 1803–1813 (66 days) with a maximum in 1810 (11 days, i.e. 3.0% of days in this year), followed by 1805 (9 days, i.e. 2,5%) and 1809 (8 days, i.e. 2.2%) (Fig. 3a). Only 14 days of missed observations occurred in 1814–1830 (with 0, 1 or 2 missing days per year, i.e. from 0 to 0.6%). As for annual distribution, the maximum of 10 missing days in February represent only 1.3% of all February days, whole 3 missing days in September correspond to 0.4% of all September days (Fig. 3b). Besides 9 missing days in August and October (1.0%), all remaining months had missing days below 1% of their days. Days with missing observations may in some way decrease presented frequencies of days with different climatic variables." Concerning of the methodology of creation of indices in 3.2 (based on your comment, the title changed on Methods of analysis), it was changed as follows from point 2 of your comments above. It means, that we combined quantitative approach (based on distribution of corresponding data and number of days with some characteristics) with an expert approach based on expert knowledge of climate in the Czech Lands. There is clear that final version of monthly indices is influenced by character of basic data, their interpretation and climatic expertise of the researcher. But having no measured data we cannot expect more. Creating the indices in three iterations, there is practically impossible to give any "descriptors producing a rank of -2, as opposed to -3" as well as "any kind of table that details terms or conditions related to each category." But we did not resign on any prove of validity of our indices series. Some tests of a quality of indices interpreted can be the high and statistically significant correlation coefficients between our indices series and measured temperatures/precipitation series in Brno (Section 5.3). Based on it we believe that our rather subjective interpretation express quite well basic features of corresponding temperature and precipitation variability (fluctuations) following from the meteorological measurements of the Brno station.

Page 6 lines 5-6-9. To what extent is this a relic of the reconstruction methodology, rather than a 'true' representation of the variability? This seems to be the same for the wetness indicators (lines 33-35) RESPONSE: Here we just described what followed from our interpretation (for its uncertainties see Section 5.1 and the fourth paragraph of Section 5.3). Each interpretation of not exactly measured values of temperatures or precipitation bears some part of subjectivity, which is difficult to quantify. We are not searching here what is a reason of this variability, we are just commenting results we obtained using data and methods of this paper. On the other hand, we believe – in case of temperatures - it is a reflection of the fact described in previous point 3 where we mentioned that the years 1803–1830 were generally cooler. Then we may expect that negative extremes could be probably better expressed than positive.

Sections 4.1.2-4 and Figures 5-7. I'm not sure about the suitability of using absolute counts of days, given that the number of recorded days is so variable. Perhaps percentages would be better? This will depend of course on the distribution of missing days – if they are relatively regularly spaced through the year this would give a stronger justification for using percentages. RESPONSE: Accepted. Fig. 6 already contains information of missing days as mentioned in the figure caption in category "6) missing days or no information related to cloudiness." Facts of missing days were already taken in account in the text in Section 4.1.3. As for Figs. 5 and 7, they were complemented directly by missing days as in Fig. 6 and in related modifications of the text in Sections 4.1.2 (the second paragraph), 4.1.4 (the first paragraph) and 4.1.5 (the first and second paragraphs) if it was necessary.

Section 5.2 and figure 9. I think this is in the wrong place in the argument, and should come in the opening sections (before the methodology). This is describing the context of the Hausner diaries, not anything within the diaries themselves. RESPONSE: Accepted. We thought that this section would be better after presentation of Hausner's observation and their analysis (i.e., the reader knows already anything about this observations), but we followed your request and shifted this section before the methodology.

Page 10 lines 15-24. Can you give any climatological suggestion for the substantially larger temperature increase to the modern period in winter compared to summer? Also, why do we see such a variation in both quantity and stdev of precipitation in August? Can this be explained by one or two years with particularly heavy rainfall during this month? RESPONSE: Accepted. The winter months are much more variable than the summer months, when particularly negative deviations are going deeper with respect to mean temperature in winter than positive deviations. Moreover, the 1803–1830 period is a part of the last phase of the LIA, characterised generally by cooler winters compared to summer. These facts should be responsible in the substantial larger temperature increase in winter than in summer. By the way, also climate models give larger changes in winter than in summer temperatures. As for August, differences between both periods clearly follow from the following data: August of the first period 1803–1830 was generally wetter (mean 69.7 mm) and less variable (standard deviation 23.9 mm, variation coefficient 35.3%), while the second period 1961–1990 was drier (mean 56.4 mm) and more variable (standard deviation 38.7 mm, variation coefficient 68.6%).

Page 10 lines 47-49. Needs a reference RESPONSE: Accepted, the reference Brázdil et al. (2012a) was added.

Page 11 – up to line 26. These comparisons are invalid, given the number of missing days in Hausner's records. Again, the use of percentages would be better, if this can be justified due to the distribution of the missing days. Also – as with the section above – can you suggest any climatological reason for the variation observed? It is true that location of the meteorological stations is likely to have an impact, but you are also looking at datasets separated by 130 years of a changing climate. RESPONSE: Accepted. But we have some doubts, that "these comparisons are invalid", because the missing values can affect corresponding monthly values only in tenths of days. For example, if in February was missing totally 10 days during 1803–1828 (Fig. 3b), it represents in average uncertainty of 0.36 days. It means that monthly values could be theoretically

underestimated maximally by values between 0.36 (February) and 0.11 (September) days. These values are not so high that should significantly change shapes of annual distribution of any analysed variables in Fig. 13. But take this comment in account, we included in the corresponding paragraph following sentence: "Because of missing Hausner's observations (see Fig. 3), his mean monthly values in Fig. 13a–d could be maximally underestimated by between 0.36 days (February) and 0.11 days (September) if in all missing days studied variables would be appearing. In case of annual values, possible maximum underestimation could achieve 3.36 days."

––––––––––––––––––––––––––––––––